# ONE PROTEIN IS ALL YOU NEED

**Anton Bushuiev**[1,2*]     **Roman Bushuiev**[1,2*]     **Olga Pimenova**[1]     **Nikola Zadorozhny**[1]
**Raman Samusevich**[1,2,8]     **Elisabet Manaskova**[1]     **Rachel Seongeun Kim**[3,4]     **Hannes Stärk**[7]
**Jiri Sedlar**[1]     **Martin Steinegger**[3,4,5,6]     **Tomáš Pluskal**[2]     **Josef Sivic**[1]

[1]Czech Institute of Informatics, Robotics and Cybernetics, Czech Technical University,
[2]Institute of Organic Chemistry and Biochemistry of the Czech Academy of Sciences,
[3]School of Biological Sciences, Seoul National University, [4]Interdisciplinary Program
in Bioinformatics, Seoul National University, [5]Institute of Molecular Biology and Genetics,
Seoul National University, [6]Artificial Intelligence Institute, Seoul National University,
[7]CSAIL, Massachusetts Institute of Technology, [8]Qminers

## ABSTRACT

Generalization beyond training data remains a central challenge in machine learning for biology. A common way to enhance generalization is self-supervised pre-training on large datasets. However, aiming to perform well on all possible proteins can limit a model's capacity to excel on any specific one, whereas experimentalists typically need accurate predictions for individual proteins they study, often not covered in training data. To address this limitation, we propose a method that enables self-supervised customization of protein language models to one target protein at a time, on the fly, and without assuming any additional data. We show that our Protein Test-Time Training (ProteinTTT) method consistently enhances generalization across different models, their sizes, and datasets. ProteinTTT improves structure prediction for challenging targets, achieves new state-of-the-art results on protein fitness prediction, and enhances function prediction on two tasks. Through two challenging case studies, we also show that customization via ProteinTTT achieves more accurate antibody–antigen loop modeling and enhances 19% of structures in the Big Fantastic Virus Database, delivering improved predictions where general-purpose AlphaFold2 and ESMFold struggle.

## 1 INTRODUCTION

A comprehensive understanding of protein structure, function, and fitness is essential for advancing research in the life sciences (Subramaniam & Kleywegt, 2022; Tyers & Mann, 2003; Papkou et al., 2023). While machine learning models have shown remarkable potential in protein research, they are typically optimized for achieving the best average performance across large datasets (Jumper et al., 2021; Watson et al., 2023; Kouba et al., 2023). However, biologists often focus their research on individual proteins or protein complexes involved in, for example, metabolic disorders (Ashcroft et al., 2023; Gunn & Neher, 2023), oncogenic signaling (Hoxhaj & Manning, 2020; Keckesova et al., 2017), neurodegeneration (Gulen et al., 2023; oh Seo et al., 2023), and other biological phenomena (Gu et al., 2022). In these scenarios, detailed insights into a single protein can lead to significant scientific advances.

**ESMFold**     **ESMFold + ProteinTTT**

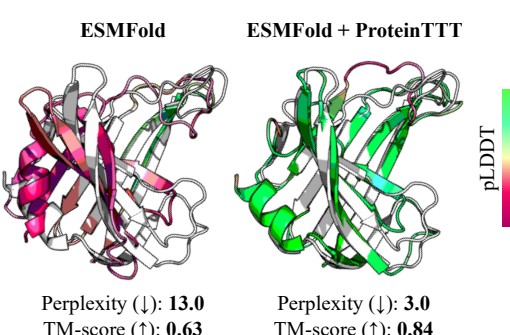

Perplexity (↓): **13.0**     Perplexity (↓): **3.0**
TM-score (↑): **0.63**     TM-score (↑): **0.84**

Figure 1: **Example of protein structure prediction after single-protein model customization via ProteinTTT.** ESMFold poorly predicts the structure of the CASP14 target T1074 (white) because the underlying language model ESM2 poorly fits the sequence, as indicated by the high perplexity (left and Fig. 2E in Lin et al. (2023)). Self-supervised test-time customization of ESM2 to the single sequence of T1074 reduces the perplexity, resulting in improved structure prediction (right).

---

[*]These authors contributed equally.

However, general machine learning models for proteins often struggle to generalize to practically interesting individual cases due to data scarcity (Bushuiev et al., 2023; Chen & Gong, 2022) and distribution shifts (Škrinjar et al., 2025; Tagasovska et al., 2024; Feng et al., 2024). Bridging the gap between broad, dataset-wide optimization and precision needed to study single proteins of practical interest remains a key challenge in integrating machine learning into biological research (Sapoval et al., 2022). This challenge is particularly acute in computational biology, where accurate predictions for individual proteins are essential to guide resource-intensive wet-lab experiments, in contrast to domains such as natural language processing or computer vision, where models are typically expected to flexibly handle diverse prompts from many users in real time (Brown, 2020; Ramesh et al., 2021).

To address this challenge, we propose a test-time approach for generalization to one protein at a time, effectively enabling more accurate predictions for individual targets, particularly those poorly represented in training data. Our Protein Test-Time Training (ProteinTTT) method customizes protein language models (PLMs) to individual proteins on the fly and without assuming additional data. Our approach is based on a simple yet powerful premise: if a language model is less perplexed (surprised) by a protein sequence–or if it "understands" its unique patterns better–it will generate a more accurate representation for predicting its structure and function. Given a model pre-trained via masked language modeling, our method effectively minimizes perplexity on a target protein or its multiple sequence alignment (MSA) through self-supervised customization, improving downstream performance without updating the downstream task head. The widespread use of masked modeling as a pre-training paradigm makes ProteinTTT broadly applicable in computational biology.

In summary, this work demonstrates the surprising effectiveness of protein model customization and lays the foundation for exploring other test-time strategies and broader biological applications. The key contributions are: **(1)** We introduce ProteinTTT, to the best of our knowledge the first customization method in machine learning for biology. We provide a user-friendly and easily extensible implementation [1] and provide insights into the effectiveness of protein model customization by linking it to perplexity minimization. **(2)** We empirically validate ProteinTTT, showing improvements in protein structure prediction with well-established models, achieving state-of-the-art results in protein fitness prediction, and enhancing protein function prediction on terpene synthase substrate classification and protein localization prediction. **(3)** We demonstrate the practical utility of focusing on one protein at a time through two challenging case studies. ProteinTTT enables more accurate prediction of antibody–antigen loops and improves 19% of structures in the Big Fantastic Virus Database, delivering accurate predictions where general-purpose AlphaFold2 and ESMFold struggle.

## 2    BACKGROUND AND RELATED WORK

The broad adoption of Y-shaped architectures relying on masked modeling enables the development of a general method for customizing protein models at test time via masking-based self-supervision.

**The Y-shaped paradigm of learning.**    In machine learning applied to proteins, architectures often follow a Y-shaped paradigm (Gandelsman et al., 2022), consisting of a backbone feature extractor $f$ operating on protein tokens $x$, a self-supervised head $g$, and an alternative fine-tuning head $h$. During training, $g \circ f$ is first pre-trained, and the pre-trained backbone $f$ is then reused to fine-tune $h \circ f$ toward a downstream task. Here, $\circ$ denotes a composition of two machine learning modules (e.g., $g$ is applied on top of $f$ in $g \circ f$). At

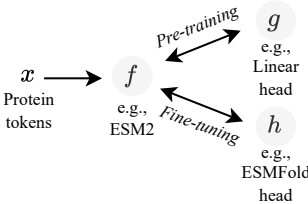

test time, the final model $h \circ f$ is fixed. Generalization is achieved by leveraging the rich knowledge encoded in the backbone $f$ and the task-specific priors embedded in the fine-tuning head $h$. This paradigm enables overcoming data scarcity during fine-tuning and underlies breakthrough approaches in protein structure prediction (Lin et al., 2023), protein design (Watson et al., 2023), protein function prediction (Yu et al., 2023), and other tasks (Hayes et al., 2024).

The backbone $f$ is typically a large neural network pre-trained in a self-supervised way on a large dataset using a smaller pre-training projection head $g$ (Hayes et al., 2024). The fine-tuning head $h$, however, depends on the application. In some cases, $h$ is a large neural network, repurposing the pre-trained model entirely (Watson et al., 2023; Lin et al., 2023); in others, $h$ is a minimal projection

[1] https://github.com/anton-bushuiev/ProteinTTT

with few parameters (Cheng et al., 2023), or even without any parameters at all (i.e., a zero-shot setup; Meier et al. (2021); Dutton et al. (2024)). The fine-tuning head $h$ can also be a machine learning algorithm other than a neural network (Samusevich et al., 2025).

**Masked modeling.** While the objective of fine-tuning $h \circ f$ is determined by the downstream application, the choice of pre-training objective for $g \circ f$ is less straightforward. Nevertheless, the dominant paradigm for protein pre-training is masked modeling, which optimizes model weights to reconstruct missing protein parts. This objective has proven effective across diverse tasks (Heinzinger & Rost, 2025; Schmirler et al., 2024), including structure (Lin et al., 2023; Jumper et al., 2021), fitness (Meier et al., 2021; Su et al., 2023), and function prediction (Samusevich et al., 2025; Yu et al., 2023; Elnaggar et al., 2021), as well as protein design (Hsieh et al., 2025; Hayes et al., 2024; Nijkamp et al., 2023), and has been successfully applied to various protein representations such as sequences (Hayes et al., 2024; Elnaggar et al., 2023), graphs (Dieckhaus et al., 2024; Bushuiev et al., 2023), and voxels (Diaz et al., 2023).

**Model customization.** Several studies have shown that machine learning models for proteins benefit from being fine-tuned on protein-specific (Notin et al., 2024; Kirjner et al., 2023; Rao et al., 2019) or protein family-specific (Sevgen et al., 2023; Samusevich et al., 2025) data. However, collecting additional data may be resource-intensive, and for many targets, relevant datasets or proteins may be limited or not available (Durairaj et al., 2023; Kim et al., 2025). In this paper, we propose a versatile method enabling customizing PLMs for a single target protein or its MSA in a self-supervised manner, on the fly, and without assuming any additional data. Customization methods have been developed in computer vision (Chi et al., 2024; Wang et al., 2023; Xiao et al., 2022; Karani et al., 2021) and natural language processing (Hübotter et al., 2024; Hardt & Sun, 2023; Ben-David et al., 2022; Banerjee et al., 2021). The paradigm of test-time training (TTT), developed to mitigate distribution shifts in computer vision applications (Gandelsman et al., 2022; Sun et al., 2020), is the main inspiration for our work. We demonstrate that customization via test-time training enhances the accuracy of PLMs across a wide range of downstream tasks even without the presence of explicit distribution shifts.

## 3 PROTEIN MODEL CUSTOMIZATION WITH PROTEINTTT

In this section, we describe the proposed Protein Test-Time Training (ProteinTTT) approach (Section 3.1), followed by its applications to a range of well-established models and datasets (Section 3.2).

### 3.1 SELF-SUPERVISED CUSTOMIZATION TO A TARGET PROTEIN

At test time, we assume a Y-shaped model with a backbone $f$ that has been pre-trained via the self-supervised track $g \circ f$, followed by task-specific fine-tuning through the supervised track $h \circ f$. The goal of customization with ProteinTTT is to adapt the backbone $f$ to a single protein $x$ before making a prediction on a downstream task via the supervised track $h \circ f$. To achieve this, we customize the backbone $f$ to the single example $x$:

$$\text{ProteinTTT} : (h \circ f(\cdot; \theta_0), x) \mapsto h \circ f(\cdot; \theta_x) \tag{1}$$

where $\theta_0$ denotes pre-trained parameters and $\theta_x$ parameters optimized for the target protein $x$ using the self-supervised track $g \circ f$, while the supervised head $h$ remains frozen. Figure 2a illustrates our customization approach, which is summarized in the following sections. Appendix C describes the extension of our method to customization using a MSA of a protein, rather than its single sequence.

**Customization training objective.** We customize $g \circ f$ to a single target protein sequence $x$ via minimizing the masked language modeling objective (Devlin, 2018; Rives et al., 2021):

$$\mathcal{L}(x; \theta) = \mathbb{E}_{M \sim p_{\text{mask}}(M)} \left[ \sum_{i \in M} -\log p(x_i | x_{\backslash M}; \theta) \right], \tag{2}$$

where $x$ denotes a sequence of protein tokens (typically amino acid types), and $\mathbb{E}_M$ represents the expectation over randomly sampled masking positions $M$. The objective function $\mathcal{L}(x; \theta)$ maximizes the log-probabilities $\log p(x_i | x_{\backslash M}; \theta) \doteq g(f(x_{\backslash M}; \theta))_i$ of the true (i.e., wild-type) tokens $x_i$ at the masked positions $i \in M$ in the partially masked sequence $x_{\backslash M}$, where $\theta$ denotes the parameters of the

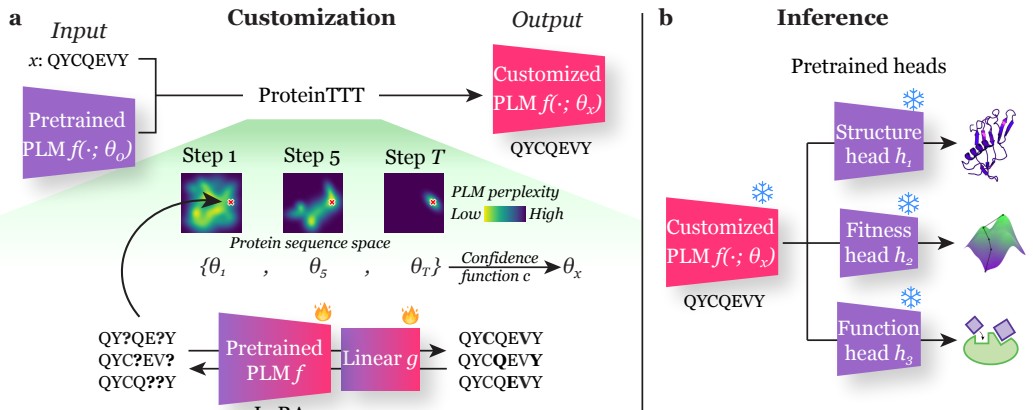

Figure 2: **Overview of protein language model (PLM) customization with ProteinTTT. (a)** Given a protein sequence of interest $x$ and a pretrained PLM $f(\cdot; \theta_0)$, ProteinTTT yields a customized version of the PLM $f(\cdot; \theta_x)$ for that sequence. Customization is achieved by fine-tuning (fire icon) the pretrained parameters $\theta_0$ via masked language modeling solely on the input sequence for $T$ steps, selecting the optimal parameters $\theta_x$ using a confidence function $c$. This procedure adapts the model specifically to the input sequence, improving its internal representation as measured by model perplexity. **(b)** Once customized, the PLM can be used with pretrained task-specific heads, such as structure, fitness, or function prediction modules, $h_1$, $h_2$, and $h_3$, respectively, without modifying their parameters (snowflake icon). For example, the ESM2 PLM can be customized and then used with the pretrained ESMFold structure prediction head without modifying its 1.4-billion task-specific parameters, resulting in improved structure prediction for the given sequence (e.g., Figure 1).

backbone $f$, and $g$ is the masked language modeling head. While we focus on classical bi-directional masked modeling, we also demonstrate that ProteinTTT can be similarly applied to autoregressive and discrete diffusion models (Appendix B).

To ensure consistency between customization and pre-training, ProteinTTT adopts the same masking and preprocessing strategies used during pre-training. Specifically, $p_{\text{mask}}(M)$ can follow different distributions, such as sampling a fixed ratio (e.g., 15%) of random tokens (Lin et al., 2023), or dynamically varying the number of sampled tokens based on another distribution (e.g., a beta distribution; Hayes et al. (2024)). During customization, we replicate the masking distribution used in pre-training. We also replicate other pre-training practices, such as replacing 10% of masked tokens with random tokens and another 10% with the original tokens (Devlin, 2018; Lin et al., 2023; Su et al., 2023) or cropping sequences to random 1024-token fragments (Lin et al., 2023; Su et al., 2023).

**Optimization.** Since customization with ProteinTTT does not assume more than a single protein, early stopping on validation data is not feasible. To address this, we first fine-tune the pre-trained parameters $\theta_0$ of a backbone $f$ for a fixed number of steps $T$, yielding parameters $\Theta = \{\theta_0, \theta_1, \ldots, \theta_T\}$. The final customized parameters $\theta_x$ are selected as $\arg\max_{\theta \in \Theta} c(h(f(x; \theta)))$ where $c$ is a confidence function. If $c$ is not available, we set $\theta_x = \theta_T$. Appendix H.2 discusses how using pLDDT as the confidence function $c$ for structure prediction makes ProteinTTT robust to hyperparameter selection and how the number of steps $T$ can be fixed (e.g., $T = 30$) while optimizing learning rate and batch size effectively. Before customizing for the next target protein, the parameters are reset to $\theta_0$.

To make ProteinTTT easily applicable to large-scale models (e.g., the 3B-parameter ESM2 backbone), we leverage low-rank adaptation (LoRA; Hu et al. (2021)) and gradient accumulation during customization. Additionally, to improve the stability and predictability of customization, we use stochastic gradient descent (SGD; Ruder (2016)) instead of the commonly used Adam optimizer (Kingma & Ba, 2015), following (Gandelsman et al., 2022). Further details are provided in Appendix F.

### 3.2 INFERENCE ON DOWNSTREAM TASKS

Once the backbone $f$ is adapted to a target protein via self-supervised customization, it can be used in conjunction with a pre-trained downstream head $h$, as $h \circ f$. The key idea of customization with

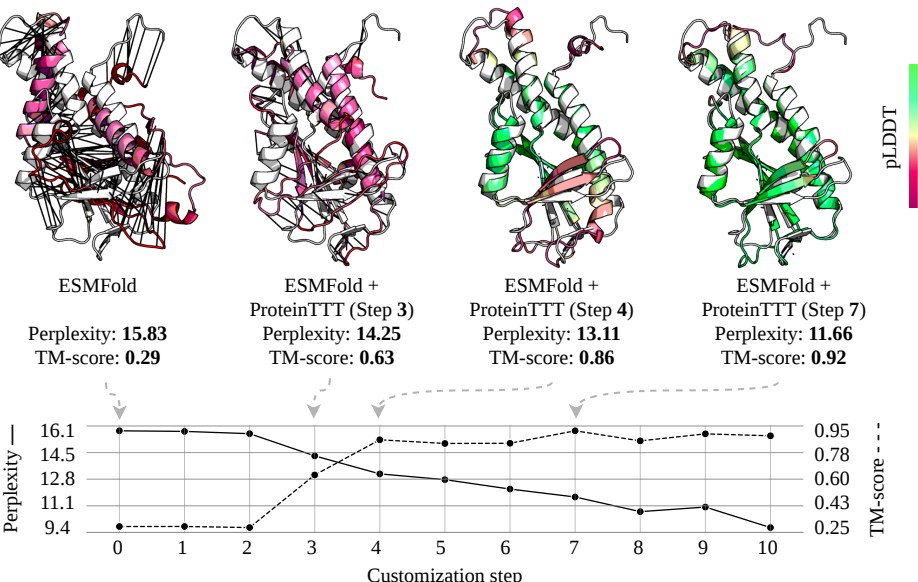

Figure 3: **Customization with ProteinTTT improves protein structure prediction by reducing protein sequence perplexity**. ESMFold fails to predict the structure of chain B from PDB entry 7EBL in the CAMEO validation set, as shown at customization step 0, where the perplexity is high and the TM-score is low. By applying customization with ProteinTTT for the single target sequence, the model iteratively improves the structure prediction quality, as demonstrated by the increasing TM-score, associated with reduced perplexity. At customization step 7, the predicted structure achieves the highest TM-score, as well as the highest predicted confidence metric pLDDT, enabling the selection of this step as the final prediction by the customized ESMFold + ProteinTTT.

ProteinTTT is not to update the head $h$, but instead to leverage improved representations from $f$ (Figure 2b). Appendix A provides a justification for why these customized representations generally enhance performance on downstream tasks by linking ProteinTTT to perplexity minimization.

Since Y-shaped architectures are prevalent in protein machine learning, ProteinTTT can be straight-forwardly applied to numerous tasks. In this work, we consider three standard problems: protein structure, fitness, and function prediction, and apply our method to corresponding well-established models. For structure prediction, we apply ProteinTTT to ESMFold (Figure 3, Lin et al. (2023), HelixFold-Single (Fang et al., 2023), DPLM2 Bit-based (Hsieh et al., 2025), and ESM3 (Hayes et al., 2024); for fitness prediction, we use ESM2 (Lin et al., 2023), SaProt (Su et al., 2023), ProSST (Li et al., 2024), MSA Transformer (Rao et al., 2021), and ProGen2 (Nijkamp et al., 2023); and for function prediction, we apply ProteinTTT to ESM-1v-based (Meier et al., 2021) EnzymeExplorer (Samusevich et al., 2025) and ESM-1b-based (Rives et al., 2021) Light attention (Stärk et al., 2021).

In all models we consider, $f$ is a Transformer encoder operating on protein tokens, and $g$ is a masked language modeling head mapping embeddings to amino acid types. The downstream head $h$, however, varies strongly by task. For structure prediction, $h$ is a structure predictor: AlphaFold2-inspired modules in ESMFold, HelixFold-Single and DPLM2 Bit-wise (Jumper et al., 2021), and a VQ-VAE decoder in ESM3 (Razavi et al., 2019). For fitness prediction, $h$ outputs a single score; all methods perform zero-shot inference using $h \circ f$ via log likelihoods from $g$, with $h$ acting as a simple, parameter-free adaptation of $g$. For function prediction, $h$ is a classifier: a random forest in EnzymeExplorer (Samusevich et al., 2025) and a light attention module in (Stärk et al., 2021).

## 4 EXPERIMENTS

In this section, we evaluate ProteinTTT on three well-established downstream tasks in protein machine learning: structure (Section 4.1), fitness (Section 4.2), and function (Section 4.3) prediction.

Table 1: **Customization with ProteinTTT improves protein structure prediction.** The metrics are averaged across 18 ESMFold low-confidence targets in the CAMEO test set, and standard deviations correspond to 5 random seeds. CoT and MP stand for the chain of thought and masked prediction baselines.

| Method | TM-score ↑ | LDDT ↑ |
|---|---|---|
| ESM3 (Hayes et al., 2024) | $0.3480_{\pm 0.0057}$ | $0.3723_{\pm 0.0055}$ |
| ESM3 + CoT (Hayes et al., 2024) | $0.3677_{\pm 0.0088}$ | $0.3835_{\pm 0.0024}$ |
| ESM3 + ProteinTTT (Ours) | $\mathbf{0.3954}_{\pm 0.0067}$ | $\mathbf{0.4214}_{\pm 0.0054}$ |
| DPLM2 Bit-based (Hsieh et al., 2025) | $0.3701_{\pm 0.0102}$ | $0.4681_{\pm 0.0071}$ |
| DPLM2 Bit-based + ProteinTTT (Ours) | $\mathbf{0.3796}_{\pm 0.0024}$ | $\mathbf{0.4742}_{\pm 0.0093}$ |
| HelixFold-Single (Fang et al., 2023) | $0.4709$ | $0.4758$ |
| HelixFold-Single + ProteinTTT (Ours) | $\mathbf{0.4839}_{\pm 0.0045}$ | $\mathbf{0.4840}_{\pm 0.0061}$ |
| ESMFold (Lin et al., 2023) | $0.4649$ | $0.5194$ |
| ESMFold + MP (Lin et al., 2023) | $0.4862_{\pm 0.0043}$ | $0.5375_{\pm 0.0070}$ |
| ESMFold + ProteinTTT (Ours) | $\mathbf{0.5047}_{\pm 0.0132}$ | $\mathbf{0.5478}_{\pm 0.0058}$ |

## 4.1 PROTEIN STRUCTURE PREDICTION

Protein structure prediction is the task of predicting 3D atom coordinates from an amino acid sequence. It is arguably one of the best-established problems in computational biology (Jumper et al., 2021).

**Evaluation setup.** To evaluate the performance of ProteinTTT, we employ CAMEO, a standard benchmark for protein folding. We use the validation and test folds from Lin et al. (2023), focusing only on targets with low-confidence predictions from the base ESMFold, as determined by pLDDT and perplexity (Appendix F.1). We use the standard TM-score (Zhang & Skolnick, 2004) and LDDT (Mariani et al., 2013) metrics to evaluate global and local structure prediction quality, respectively.

As baseline methods, we use techniques alternative to ProteinTTT for improving the performance of the pre-trained base models. In particular, the ESMFold paper proposes randomly masking 15% of amino acids in a protein sequence before inference, allowing for sampling multiple protein structure predictions from the regression ESMFold model (Lin et al., 2023). For each sequence, we sample a number of predictions equal to the total number of ProteinTTT steps and refer to this baseline as ESMFold + MP (Masked Prediction). As a baseline for ESM3, we use chain-of-thought iterative decoding, referred to as ESM3 + CoT, proposed in the ESM3 paper (Hayes et al., 2024).

**Results.** Customization with ProteinTTT consistently improves the performance of all the tested methods, ESMFold, HelixFold-Single, and ESM3, outperforming the masked prediction (ESMFold + MP) and chain-of-thought (ESM3 + CoT) baselines, as shown in Table 1. Among the 18 challenging CAMEO test proteins, ProteinTTT significantly improved the prediction of 7, 4, 5, and 6 structures from ESMFold, DPLM2 Bit-based, HelixFold-Single, and ESM3, respectively, while only moderately disrupting the prediction of 2, 1, 1, and 1 structures, respectively (Figure A6). Remarkably, ProteinTTT improves DPLM2 Bit-based despite the absence of a confidence function (no trained pLDDT head available) and despite the model being pretrained via discrete diffusion, while still using the same masked-modeling objective for customization as for the other methods.

Most notably, ProteinTTT enables accurate structure prediction for targets that are poorly predicted with the original models. For instance, Figure 1 presents a strongly improved structure predicted using ESMFold + ProteinTTT for the target that was part of the CASP14 competition and shown as an unsuccessful case in the original ESMFold publication (Lin et al. (2023), Fig. 2E). Another example is shown in Figure 3, where ProteinTTT refined the structure prediction from a low-quality prediction (TM-score = 0.29) to a nearly perfectly folded protein (TM-score = 0.92). Figure A4 shows that ESMFold + ProteinTTT maintains computational efficiency of ESMFold, being an order of magnitude faster than AlphaFold2. Figure A11 additionally demonstrates the robustness of ESM3 + ProteinTTT to the choice of hyperparameters.

Table 2: **Customization with ProteinTTT improves protein fitness prediction.** The right section of the table presents performance averaged across individual proteins and then across different protein phenotypes, as classified in the ProteinGym benchmark (Notin et al., 2024). The middle column shows the final performance, averaged across all five phenotype classes. In total, ProteinGym contains 2.5 million mutations across 186 proteins. Standard deviations are calculated over 5 random seeds.

| | Avg. Spearman ↑ | Spearman by phenotype ↑ | | | | |
| --- | --- | --- | --- | --- | --- | --- |
| | | Activity | Binding | Expression | Organismal Fitness | Stability |
| ESM2 (35M) (Lin et al., 2023) | 0.3211 | 0.3137 | 0.2907 | 0.3435 | 0.2184 | 0.4392 |
| ESM2 (35M) + ProteinTTT (Ours) | **0.3407** ± 0.00014 | **0.3407** | **0.2942** | **0.3550** | **0.2403** | **0.4733** |
| ProGen2-small (151M) (Nijkamp et al., 2023) | 0.3255 | 0.3316 | 0.2681 | 0.3730 | 0.3283 | 0.3264 |
| ProGen2-small (151M) + ProteinTTT (Ours) | **0.3591** ± 0.00021 | **0.3827** | **0.2960** | **0.3875** | **0.3302** | **0.3992** |
| ProGen2-large (2.7B) (Nijkamp et al., 2023) | 0.3724 | 0.4001 | 0.2820 | 0.4184 | **0.3711** | 0.3904 |
| ProGen2-large (2.7B) + ProteinTTT (Ours) | **0.3817** ± 0.00158 | **0.4091** | **0.3113** | **0.4227** | 0.3661 | **0.3993** |
| SaProt (35M) (Su et al., 2023) | 0.4062 | 0.3721 | 0.3568 | 0.4390 | 0.2879 | 0.5749 |
| SaProt (35M) + ProteinTTT (Ours) | **0.4106** ± 0.00004 | **0.3783** | **0.3569** | **0.4430** | **0.2955** | **0.5795** |
| ESM2 (650M) (Lin et al., 2023) | 0.4139 | 0.4254 | 0.3366 | 0.4151 | 0.3691 | **0.5233** |
| ESM2 (650M) + ProteinTTT (Ours) | **0.4153** ± 0.00003 | **0.4323** | **0.3376** | **0.4168** | **0.3702** | 0.5195 |
| SaProt (650M) (Su et al., 2023) | 0.4569 | 0.4584 | 0.3785 | **0.4884** | 0.3670 | **0.5919** |
| SaProt (650M) + ProteinTTT (Ours) | **0.4583** ± 0.00001 | **0.4593** | **0.3790** | 0.4883 | **0.3754** | 0.5896 |
| ProSST (K=2048) (Li et al., 2024) | 0.5068 | 0.4758 | 0.4448 | 0.5302 | 0.4306 | **0.6526** |
| ProSST (K=2048) + ProteinTTT (Ours) | **0.5087** ± 0.00004 | **0.4822** | **0.4470** | **0.5321** | **0.4315** | 0.6507 |

## 4.2 PROTEIN FITNESS PREDICTION

The task of protein fitness prediction is to accurately order mutations of a protein based on their disruptive/favorable effects on protein functioning.

**Evaluation Setup.** We evaluate the models using ProteinGym, the state-of-the-art fitness prediction benchmark (Notin et al., 2024), focusing on its well-established zero-shot setup. Since the zero-shot setup only provides a test set without any data split, we also validate ProteinTTT on independent data. To achieve this, we create a new fitness prediction dataset mined from MaveDB, a public repository of Multiplexed Assays of Variant Effect (MAVEs) (Esposito et al., 2019). Following ProteinGym, we report Spearman correlation between predicted and experimental fitness values.

**Results.** ProteinTTT consistently enhances fitness prediction performance of all the tested models across varying model scales (35M and 650M parameters for both ESM2 and SaProt; 110M for ProSST) and both datasets, i.e., test ProteinGym (Table 2) and validation MaveDB (Table A5). Notably, ProSST + ProteinTTT sets a new state of the art on the ProteinGym benchmark (Spearman correlation coefficients calculated for individual deep mutational scanning experiments (DMSs) have statistically significant difference according to a paired t-test with $p < 0.05$).

We observe that ProteinTTT primarily improves performance for proteins with low MSA depth (i.e., the number of available homologous sequences), suggesting that single-sequence customization enhances predictions for proteins with fewer similar sequences in the training data (Table A4). The fact that ProteinTTT more effectively improves the performance of smaller ESM2 and SaProt models compared to their larger variants may be a result of the benchmark performance being saturated for larger models, consistent with a recent observation (Notin, 2025). We provide a qualitative example showing how ESM2 (650M) + ProteinTTT significantly improves fitness prediction by capturing residues critical for protein stability (Figure A5). We also demonstrate that customization can be combined with evolutionary information from MSA to further boost fitness prediction (Appendix C).

## 4.3 PROTEIN FUNCTION PREDICTION

Finally, we demonstrate a proof of concept for customization in the context of protein function prediction. We experiment with two tasks: predicting protein location within a cell (Stärk et al., 2021), and substrate classification for terpene synthases (TPS), enzymes producing the largest class of natural products (Samusevich et al., 2025). Appendix D shows that per-protein customization with ProteinTTT consistently enhances the performance of representative models on both tasks.

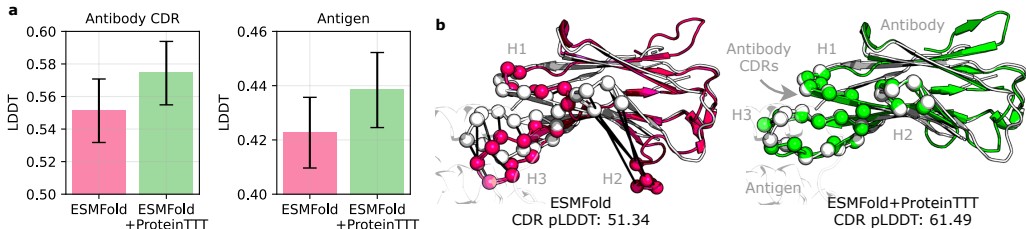

Figure 4: **ProteinTTT improves modeling of antibody–antigen loops**. **(a)** Average LDDT on the antibody complementarity-determining regions (CDRs, 175 structures) and antigens (814 structures) from the SAbDab dataset with ESMFold pLDDT < 70. Error bars indicate 95% confidence intervals estimated from 1000 bootstrap samples. **(b)** Example of improved structure prediction for CDRs in the 8K2W entry. The CDR regions H1, H2, and H3, i.e., the parts of the antibody that bind to the antigen, are highlighted with spheres, while black lines show the alignment error between the ground-truth CDR structure (white) and the predictions (colored).

## 5 CASE STUDIES

ProteinTTT can be incorporated into structure, fitness, or function prediction pipelines with a few lines of code (Appendix E). Here, we demonstrate two challenging case studies: improving modeling of antibody–antigen loops (Section 5.1) and expanding known structures of viral proteins (Section 5.2).

### 5.1 MODELING ANTIBODY–ANTIGEN LOOPS

Accurately predicting structures of antibodies (e.g., human defensive proteins) and antigens (e.g., viral proteins) enables rational design of new therapeutics (Bennett et al., 2025). However, the presence of highly variable loop regions makes modeling of these interactions a long-standing challenge. Here, we show that ProteinTTT substantially improves structure prediction for these loop-formed complementarity-determining regions (CDRs) of antibodies, i.e., the parts that bind antigens, as well as for antigens themselves, on the well-established SAbDab dataset (Dunbar et al., 2014).

We take the structures from SAbDab that are not predicted well by ESMFold (pLDDT < 70) and show that ProteinTTT improves the LDDT score for 115 of 175 antibody CDR substructures (66%) and 487 of 814 antigen chains (60%). As shown in Figure 4a, ESMFold + ProteinTTT achieves significantly higher average LDDT scores compared to general-purpose ESMFold (paired t-test p-value < 0.05). Figure 4b illustrates how ProteinTTT enables accurate prediction of all three CDRs in an antibody chain, providing an improved understanding of its binding interface with the corresponding antigen.

### 5.2 EXPANDING KNOWN STRUCTURES OF VIRAL PROTEINS

Predicting the structures of viral proteins is vital for vaccine development, antiviral design, and understanding infection (Bravi, 2024). Nevertheless, it remains challenging due to the high mutation rate, which often leaves viral proteins without close homologs or experimental structures in databases (Kim et al., 2025). Here, we demonstrate that per-protein customized predictions with ESMFold + ProteinTTT improve viral protein structure prediction, substantially expanding the Big Fantastic Virus Database—the comprehensive repository of 351,242 viral protein structures (Kim et al., 2025).

Among all the entries in BFVD, predicted with AlphaFold2 through ColabFold (Mirdita et al., 2022) using MSAs constructed from Logan (Chikhi et al., 2024), only 55% have high-quality structure predictions (pLDDT > 70). We apply ESMFold and ESMFold + ProteinTTT to the BFVD entries to expand the database with higher-quality structures. This is achieved by applying all three methods to the specific protein and taking the predicted structure with the highest pLDDT. While ESMFold manages to improve the predicted structure (as measured by pLDDT) for 10% of the BFVD proteins, ESMFold + ProteinTTT leads to an improvement for 19% of the dataset entries, substantially increasing the quality of known viral protein structures (Figure 5a).

We validate that the improved pLDDT confidence values from ESMFold + ProteinTTT correlate with the quality of the predicted structures, as measured by LDDT against reference AlphaFold2 structures having pLDDT > 90 (Pearson = 0.875; Figure A9). Notably, the largest improvements

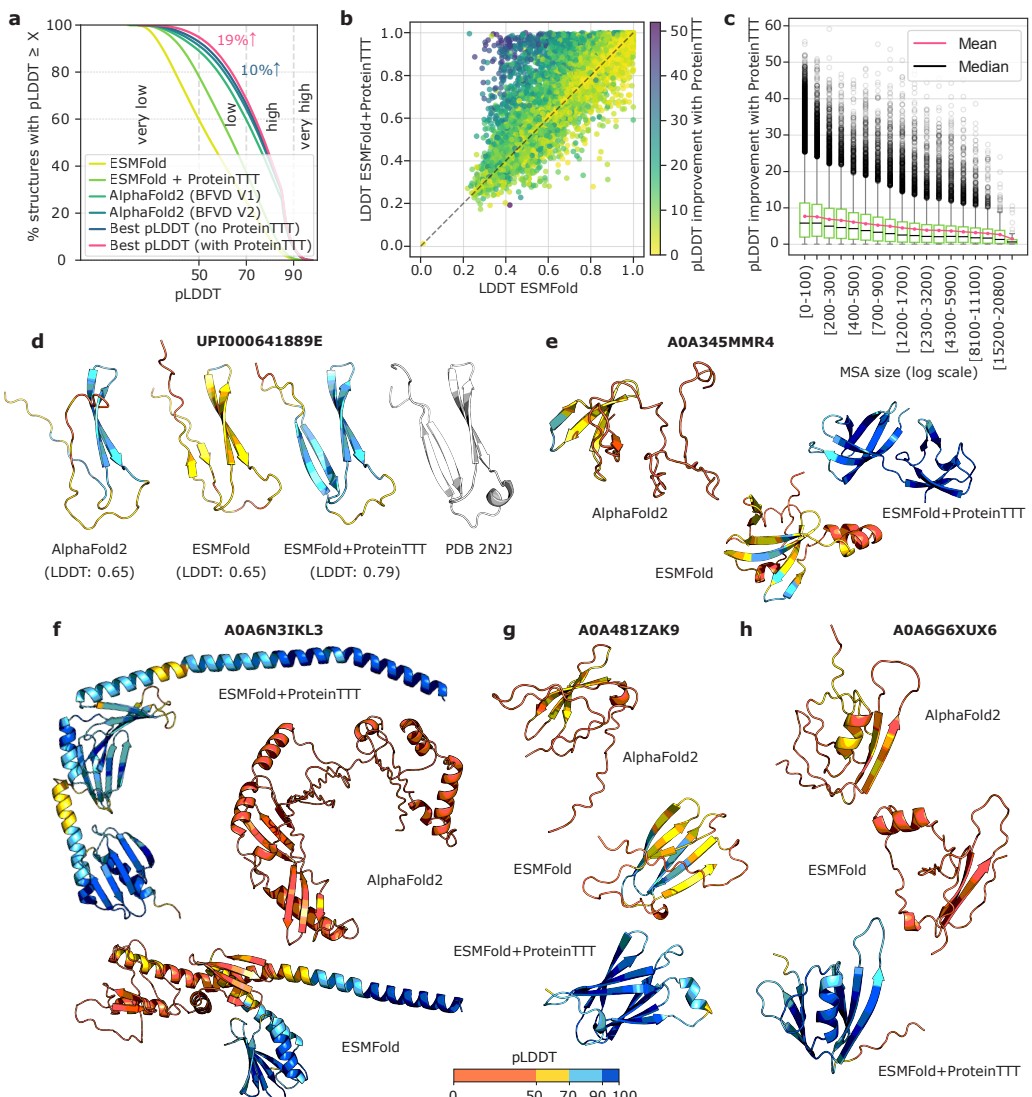

Figure 5: **ProteinTTT expands the Big Fantastic Virus Database (BFVD). (a)** ProteinTTT (light green) substantially improves the performance of ESMFold (yellow) on viral proteins, yielding better structures (pink) for 19% of BFVD entries compared to the original predictions by AlphaFold2 (green). **(b)** Improvements in pLDDT for ESMFold after ProteinTTT correspond to improvements in LDDT, as benchmarked against BFVD AlphaFold2 structures with pLDDT > 90. **(c)** ProteinTTT provides the largest pLDDT improvements (y-axis) for the most out-of-distribution proteins, i.e., those with the smallest MSAs (left on the x-axis) from the Logan database. **(d)** Structural comparison for BFVD entry UPI000641889E against the PDB structure 2N2J (100% sequence identity) shows that ESMFold + ProteinTTT yields a prediction closest to the ground truth (gray), as also measured by LDDT. **(e–g)** Additional examples of high-quality viral structures (as measured by pLDDT) predicted with ESMFold + ProteinTTT but not with ESMFold or AlphaFold2. Higher pLDDT values are better.

in pLDDT align with the largest improvements in LDDT (Figure 5b). We find that the benefit of customization saturates with the number of homologs available for a protein, indicating that ProteinTTT is most effective for challenging, out-of-distribution proteins (Figure 5c). Finally, Figure 5d–g shows examples where ProteinTTT enables high-confidence structure predictions in cases where general-purpose, uncustomized AlphaFold2 and ESMFold struggle.

## 6 DISCUSSION

We introduce ProteinTTT, a method for customizing protein language models to individual targets. ProteinTTT consistently improves performance across various models, their scales, and downstream tasks. It excels on challenging, out-of-distribution examples where general models often fail. We demonstrate its practical value through two case studies: enhancing the structural prediction of difficult antibody-antigen loops and improving 19% of low-confidence viral protein structures in the Big Fantastic Virus Database. Our work establishes per-protein customization as a powerful and practical tool for biological research.

### ACKNOWLEDGMENTS

We thank Milot Mirdita for discussions and feedback about this work. This work was supported by the Ministry of Education, Youth and Sports of the Czech Republic through projects e-INFRA CZ [ID:90254], ELIXIR [LM2023055], CETOCOEN Excellence CZ.02.1.01/0.0/0.0/17_043/0009632, ESFRI RECETOX RI LM2023069. This work was also supported by the CETOCOEN EXCELLENCE Teaming project supported from the European Union's Horizon 2020 research and innovation programme under grant agreement No 857560. This work was also supported by the European Union, ERC FRONTIER (No. 101097822), ELIAS (No. 101120237)) and the Technology Agency of the Czech Republic under the NCC Programme from the state budget (No. RETEMED TN02000122) and under the NRP from the EU RRF (No. TEREP TN02000122/001N). This work was also supported by the EU's Horizon Europe Programme under the Grant agreement No. 101136607 (CLARA Project), and was co-funded by the EU from the Operational Programme Jan Amos Komenský (OP JAK) (project "Center for Artificial Intelligence and Quantum Computing in System Brain Research, reg. no. CZ.02.01.01/00/23_029/0008437). This work was also supported by the Czech Science Foundation (GA CR) grant 21-11563M and by the European Union's Horizon Europe program (ERC, TerpenCode, 101170268). Views and opinions expressed are however those of the author(s) only and do not necessarily reflect those of the European Union or the European Research Council. Neither the European Union nor the granting authority can be held responsible for them. M.S. acknowledges support by the National Research Foundation of Korea grants (RS-2020-NR049543, RS-2021-NR061659 and RS-2021-NR056571, RS-2024-00396026), Creative-Pioneering Researchers Program and Novo Nordisk Foundation (NNF24SA0092560). This work was supported with funding from Google.org via the Google PhD Fellowship (R.B.).

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

# APPENDIX

## CONTENTS

## A  JUSTIFICATION OF CUSTOMIZATION VIA PERPLEXITY MINIMIZATION

While the paradigm of test-time customization has been investigated in other domains, the reasons behind its surprising effectiveness are not completely clear (Liu et al., 2021; Zhao et al., 2023). Here, we offer a potential justification for the effectiveness of ProteinTTT by linking it to perplexity minimization.

Perplexity has traditionally been used in natural language processing to evaluate how well models comprehend sentences (Brown, 2020; Chelba et al., 2013). Protein language modeling has adopted this metric to assess how effectively models "understand" amino acid sequences (Hayes et al., 2024; Lin et al., 2023). For bidirectional, random masking language models, which are the focus of this study, we consider the following definition of perplexity[2]:

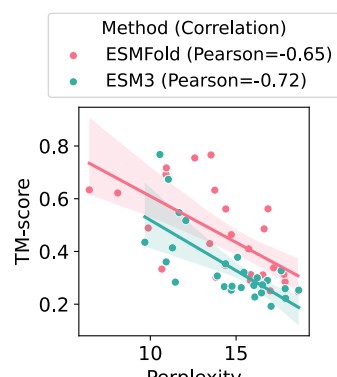

Figure A1: Quality of protein structure prediction, as measured by TM-score, correlates with perplexity of the underlying language model on the challenging targets from the CAMEO validation set. Higher TM-scores are associated with lower perplexity, indicating that better predictions are linked to lower uncertainty in the language model's understanding of the protein sequence.

$$\text{Perplexity}(x) = \exp\left(\frac{1}{|x|}\sum_{i=1}^{|x|} -\log p(x_i|x_{\setminus i};\theta)\right), \quad (3)$$

where $|x|$ is the length of the input protein sequence $x$ and $p(x_i|x_{\setminus i};\theta)$ represents the probability that the model correctly predicts the token $x_i$ at position $i$ when it is masked on the input $x_{\setminus i}$. Perplexity ranges from 1 to infinity (the lower, the better), providing an intuitive measure of how well a model fits, on average, tokens in a given sequence. A perplexity value of 1 indicates that the model perfectly fits the sequence, accurately predicting all the true tokens.

Several studies have shown that lower perplexity on held-out protein sequences (calculated through the self-supervised track $g \circ f$) correlates with better performance on downstream tasks (via the supervised track $h \circ f$), such as predicting protein contacts (Rao et al., 2020), structure (Lin et al., 2023), or fitness (Kantroo et al., 2024). To give an example, we analyze the correlation between perplexity and structure prediction quality (Figure A1; see Section 4.1 for experimental details). A notable correlation suggests that reducing a model's perplexity on a single target sample $x$ (applied independently to all test samples) can lead to improved predictions on the downstream task (Figure 3; Figure A10).

Since we assume only a single target example $x$, the minimization of the masked language modeling loss $\mathcal{L}(x;\theta)$ (Equation (2)) on this example is directly linked to minimizing the perplexity Perplexity$(x)$ (Equation (3)). For instance, in the case of a single masked position (i.e., $|M| = 1$), the loss is equal to the logarithm of perplexity. More generally, it can be shown formally that by minimizing the masked language modeling objective, the model learns to approximate the conditional marginals of the language (of proteins), including the leave-one-out probabilities evaluated in perplexity (Hennigen & Kim, 2023). As a result, applying self-supervised test-time customization on $x$ through $g \circ f$ enhances the representation of the target protein in the backbone $f$, leading to improved downstream performance via the fine-tuning track $h \circ f$.

## B  CUSTOMIZATION BEYOND MASKED LANGUAGE MODELING

In this work, we primarily focus on protein language models pretrained with masked language modeling (MLM), where a fixed proportion of randomly selected tokens (e.g., 15%) are masked for training. To date, MLM has been the dominant paradigm in protein representation learning. Nevertheless, we also provide a proof of concept showing that ProteinTTT can be applied to autoregressive and discrete

---

[2]Please note that this is an approximation of perplexity, which is computationally intractable for bidirectional models, and is often referred to as pseudo-perplexity (Lin et al., 2023; Salazar et al., 2019).

diffusion–based protein language models, with details provided in the corresponding paragraphs below. Furthermore, in Appendix I we discuss how ProteinTTT could be extended beyond protein language models.

**Autoregressive customization objective.** To perform single-sequence customization in an autoregressive setting (i.e., customization of ProGen2 (Nijkamp et al., 2023)), we apply a standard teacher forcing procedure (Vaswani, 2017) with a batch size of one. Specifically, each ProteinTTT step optimizes next token prediction across the whole sequence in parallel via the following loss function:

$$\mathcal{L}_{\text{AR}}(x;\theta) = \frac{1}{|x|} \sum_{i=1}^{|x|} -\log p(x_i \mid x_{<i};\theta), \tag{4}$$

where $x$ denotes a sequence of protein tokens, and $p(x_i \mid x_{<i};\theta) \doteq g(f(x_{<i};\theta))_{x_i}$ is the probability assigned by the model to the true token $x_i$ given all preceding tokens $x_{<i}$. Here, we use the notation consistent with Equation (2).

**Discrete diffusion customization objective.** Recently, discrete diffusion protein language models have emerged as an extension of MLM-based protein language models. Instead of masking a fixed ratio of tokens, discrete diffusion approaches vary the masking ratio during training according to a diffusion schedule (Hsieh et al., 2025; Wang et al., 2024b;a; Campbell et al., 2024; Alamdari et al., 2023). This has been shown to improve representation learning and to enable sequence generation by starting from a fully masked sequence and gradually denoising it (Wang et al., 2024a).

In this work, we experiment with the DPLM2 Bit-based discrete diffusion model (Hsieh et al., 2025) for protein structure prediction. Interestingly, we find that using a standard MLM objective with a fixed 15% masking ratio for customization (Equation (2)) already improves performance. Exploring modifications of the customization objective tailored specifically to discrete diffusion models presents an exciting direction for future work.

## C    CUSTOMIZATION WITH MULTIPLE SEQUENCE ALIGNMENT (MSA)

Table A1: **ProteinTTT can be used with MSA when available.** Please see Table 2 for evaluation details.

| Method | Avg. Spearman ↑ |
|---|---|
| ESM2 (Lin et al., 2023) | 0.4139 |
| ESM2 + ProteinTTT$_{\text{MSA}}$ (Ours) | **0.4299** $_{\pm\,0.00099}$ |
| MSA Transformer (Rao et al., 2021) | 0.4319 |
| MSA Transformer + ProteinTTT (Ours) | **0.4326** $_{\pm\,0.00003}$ |

**Customization training objective.** Since many target proteins may not have homologous sequences (Rao et al., 2021) and finding such homologs may be time-consuming (Lin et al., 2023), the ProteinTTT customization objective (Equation (2)) only assumes a single target sequence for customization. However, we also extend the loss function to the case when a multiple sequence alignment (MSA) is available:

$$\mathcal{L}_{\text{MSA}}(x;\theta) = \mathbb{E}_{x' \sim p_{\text{MSA}}(x'|x)} \big[ \mathcal{L}(x';\theta) \big], \tag{5}$$

where $p_{\text{MSA}}(x'|x)$ is the distribution of sequences $x'$ homologous to the target protein $x$, $\mathcal{L}$ is the single-sequence loss function defined in Equation (2), and $\theta$ denotes the tunable parameters of the model backbone $f$. We refer to customization using Equation (5) as ProteinTTT$_{\text{MSA}}$.

**Results for fitness prediction.** It is known that evolutionary information is important for protein fitness prediction (Laine et al., 2019). Therefore, we demonstrate how ProteinTTT$_{\text{MSA}}$ and ProteinTTT can enhance the performance of PLMs on the ProteinGym benchmark (Notin et al., 2024). Table A1 shows that using ProteinTTT$_{\text{MSA}}$ with high-quality MSAs curated by Notin et al. (2024) strongly enhances the performance of ESM2, approaching that of MSA Transformer, pre-trained on MSAs. Moreover, we find that MSA Transformer slightly benefits from single-sequence customization

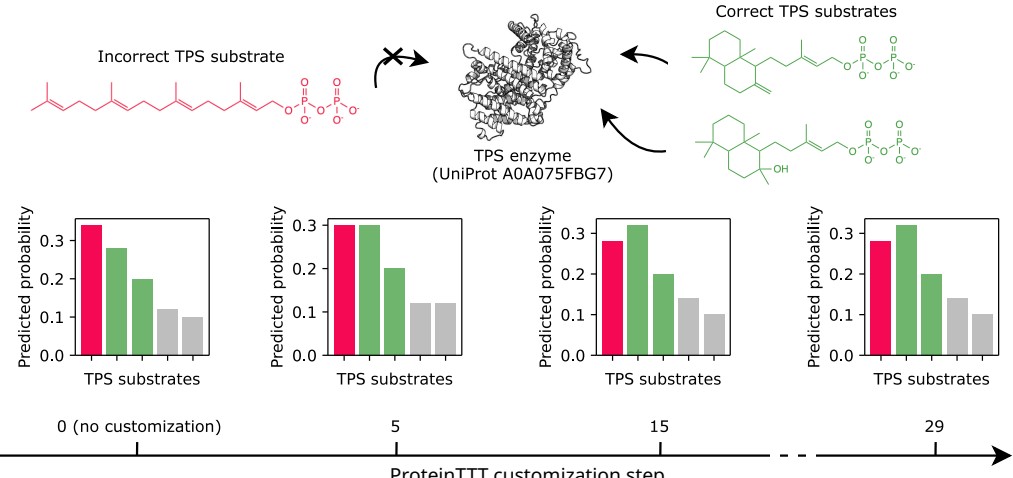

Figure A2: **Customization with ProteinTTT enables the correct substrate classification for a terpene synthase (TPS) enzyme.** With progressive customization steps of EnzymeExplorer + ProteinTTT, the probability of the initially misclassified substrate (red) decreases, while the probability of the true substrates (green) increases. The bar plots also display the predicted probabilities for other substrates with non-zero values (grey).

with ProteinTTT, while customization to whole or subsampled MSAs disrupts the performance (Table A3 in Appendix H.2). Please note that similar results were previously demonstrated in (Gordon et al., 2024) and (Alley et al., 2019) by fine-tuning protein language models on MSA, referred to as "evotuning".

## D    CUSTOMIZATION FOR PROTEIN FUNCTION PREDICTION

Protein function prediction is essential for understanding biological processes and guiding bioengineering, but is challenging due to its vague definition and limited data (Yu et al., 2023; Radivojac & et al., 2013; Stärk et al., 2021; Mikhael et al., 2024; Samusevich et al., 2025). While improved structure prediction with ProteinTTT (Section 4.1) can already enhance function prediction (Song et al., 2024), we also evaluate our customization method directly on two function classification tasks: subcellular localization, predicting protein location within a cell (Stärk et al., 2021), and substrate classification for terpene synthases (TPS), enzymes producing the largest class of natural products (Christianson, 2017; Samusevich et al., 2025). Using ProteinTTT with EnzymeExplorer (Samusevich et al., 2025) for TPS detection and Light attention (Stärk et al., 2021) for subcellular localization, we achieve consistent performance gains.

**Evaluation setup.**    For the terpene substrate classification, we use the largest available dataset of characterized TPS from Samusevich et al. (2025) and reuse the original cross-validation schema. In the case of protein localization prediction, we use a standard DeepLoc dataset (Almagro Armenteros et al., 2017) as a validation set and setHard from Stärk et al. (2021) as the test set.

Given a protein, the goal of function prediction is to correctly classify it into one of the predefined functional annotations. We assess the quality of the TPS substrate prediction using standard multi-label classification metrics used in the EnzymeExplorer paper (Samusevich et al., 2025): mean average precision (mAP) and area under the receiver operating characteristic curve (AUROC). In the case of protein localization prediction, we similarly use the classification metrics from the original paper (Stärk et al., 2021): accuracy, multi-class Matthews correlation coefficient (MCC), and F1-score.

**Results.**    Customization with ProteinTTT improves model performance on both of the protein function prediction tasks and across all considered metrics (Table A2). Figure A2 provides a qualitative

Table A2: **Customization with ProteinTTT improves protein function prediction.** For the terpene syntase (TPS) substrate classification task, the metrics are computed on the 512 TPS sequences based on the cross-validation schema of the TPS dataset (Samusevich et al., 2025). Subcellular localization prediction performance is reported for 432 protein sequences from the setHard test set (Stärk et al., 2021). The error bars show standard deviations across five random seeds.

| TPS substrate classification | | |
| --- | --- | --- |
| Method | mAP ↑ | AUROC ↑ |
| EnzymeExplorer (Samusevich et al., 2025) | 0.805 | 0.948 |
| EnzymeExplorer + ProteinTTT (Ours) | **0.811 ± 0.0011** | **0.950 ± 0.0002** |

| Subcellular localization prediction | | | |
| --- | --- | --- | --- |
| Method | Accuracy ↑ | MCC ↑ | F1-score ↑ |
| Light attention (Stärk et al., 2021) | 0.627 | 0.549 | 0.618 |
| Light attention + ProteinTTT (Ours) | **0.634 ± 0.004** | **0.557 ± 0.005** | **0.627 ± 0.004** |

result, where customization with ProteinTTT iteratively refines the prediction of EnzymeExplorer toward a correct TPS substrate class. We hypothesize that improvement with customization is more challenging in classification tasks, as opposed to regression problems, because a larger change in the latent space is required to shift the top-class probability.

# E IMPLEMENTATION DETAILS

```python
import esm
from proteinttt.models.esmfold import ESMFoldTTT, DEFAULT_ESMFOLD_TTT_CFG

# Set protein sequence
sequence = (
    "GIHLGELGLLPSTVLAIGYFENLVNIICESLNMLPKLEVSGKEYKKFKFTIVIPKDLDANIKKRAKIY"
    "FKQKSLIEIEIPTSSRNYPIHIQFDENSTDDILHLYDMPTTIGGIDKAIEMFMRKGHIGKTDQQKLLE"
    "ERELRNFKTTLENLIATDAFAKEMVEVIIEE"
)

# Load model
model = esm.pretrained.esmfold_v1()
model = model.eval().cuda()

predict_structure(model, sequence)
# pLDDT: 38.43025

# =========================== ProteinTTT ================================
# Customize model to sequence
model = ESMFoldTTT.ttt_from_pretrained(
    model, ttt_cfg=DEFAULT_ESMFOLD_TTT_CFG, esmfold_config=model.cfg
)
model.ttt(sequence)
# =====================================================================

predict_structure(model, sequence)
# pLDDT: 78.69619

# =========================== ProteinTTT ================================
# Reset model to original state (after this model.ttt can be called with
# another protein)
model.ttt_reset()
# =====================================================================
```

Code snippet 1: Incorporation of ProteinTTT into an ESMFold structure prediction pipeline using the `proteinttt` package.

```python
import torch
import esm
from esm.model.esm2 import ESM2
from proteinttt.base import TTTModule

class ESM2TTT(TTTModule, ESM2):
    def __init__(self, ttt_cfg: TTTConfig, **kwargs):
        ESM2.__init__(self, **kwargs)
        TTTModule.__init__(self, ttt_cfg=ttt_cfg)
        self.ttt_alphabet = esm.Alphabet.from_architecture("ESM-1b")
        self.ttt_batch_converter = self.ttt_alphabet.get_batch_converter()

    def _ttt_tokenize(self, seq: str, **kwargs):
        batch_labels, batch_strs, batch_tokens = self.ttt_batch_converter(
            [(None, seq)]
        )
        return batch_tokens

    def _ttt_get_frozen_modules(self) -> list[torch.nn.Module]:
        return [self.embed_tokens]

    def _ttt_mask_token(self, token: int) -> int:
        return self.ttt_alphabet.mask_idx

    def _ttt_get_padding_token(self) -> int:
        return self.ttt_alphabet.padding_idx

    def _ttt_token_to_str(self, token: int) -> str:
        return self.ttt_alphabet.all_toks[token]

    def _ttt_get_all_tokens(self) -> list[int]:
        return [
            self.ttt_alphabet.tok_to_idx[t]
            for t in self.ttt_alphabet.all_toks
        ]

    def _ttt_get_non_special_tokens(self) -> list[int]:
        return [
            self.ttt_alphabet.tok_to_idx[t]
            for t in self.ttt_alphabet.standard_toks
        ]

    def _ttt_predict_logits(
        self, batch: torch.Tensor, start_indices: torch.Tensor = None
    ) -> torch.Tensor:
        return self(batch)["logits"]
```

Code snippet 2: Implementation of ESM2 + ProteinTTT within the `proteinttt` package.

**Infrastructure.** All experiments with ProteinTTT are conducted on machines equipped with a single NVIDIA A100 40GB GPU, an 8-core AMD processor, and 128 GB of physical memory.

**Source code.** We provide a user-friendly and easily extensible PyTorch (Paszke, 2019) implementation of ProteinTTT, available as the `proteinttt` Python package [3]. We provide Code snippet 1 and Code snippet 2 in Python to demonstrate the implementation of inference and customization with ProteinTTT, respectively. Code snippet 1 demonstrates how inference with ESMFold can be enhanced with ProteinTTT by adding just a few lines of code to enable customization. Next, Code snippet 2 shows how ProteinTTT can be easily implemented for a PLM of interest by inheriting from the abstract `TTTModule` class. To integrate ProteinTTT within a model (e.g., ESM2), the user needs

---

[3] https://github.com/anton-bushuiev/ProteinTTT

to implement methods that define the model's vocabulary, an interface for predicting logits, and a specification of which modules need to be fine-tuned or remain frozen. The rest, i.e., the test-time training logic itself, is implemented within the unified `TTTModule` class.

**Optimization.** We minimize the loss defined in Equation (2) using stochastic gradient descent (SGD) with zero momentum and zero weight decay (Ruder, 2016). While a more straightforward option might be to use the optimizer state from the final pre-training step, this approach is often impractical because the optimizer parameters are usually not provided with the pre-trained model (Hayes et al., 2024; Lin et al., 2023). Moreover, many models are pre-trained using the Adam optimizer (Kingma & Ba, 2015) or its variants (Loshchilov & Hutter, 2019). However, it was shown that Adam results in less predictable behavior of test-time training compared to the SGD optimizer, possibly due to its more exploratory behavior (Gandelsman et al., 2022).

**Customizing large models.** We aim for customization to be applicable on the fly, i.e., without the need for any pre-computation and on a single GPU with a minimum computational overhead. Since state-of-the-art models for many protein-oriented tasks are typically large, with up to billions of parameters, our aim presents two key challenges. First, when using pre-trained Transformers on a single GPU, even for the forward pass, the batch size is typically limited to only several samples due to the quadratic complexity of the inference (Vaswani, 2017). Second, for the backward pass, even a batch size of one is not always feasible for large models. To address the first challenge, we perform forward and backward passes through a small number of training examples and accumulate gradients to simulate updates with any batch size. We address the second challenge by employing low-rank adaptation (LoRA; Hu et al. (2021)), which in practice enables fine-tuning of any model for which a forward pass on a single sample is feasible, due to a low number of trainable parameters. Appendix H.3 details how ESMFold (Lin et al., 2023), with its 3B-parameter ESM2 backbone $f$, can be efficiently customized, retaining its speed advantage while enhancing performance.

## F    EXPERIMENTAL DETAILS

In this section, we describe the proposed benchmark suite for the three customization tasks considered in this work: protein structure prediction (Appendix F.1), protein fitness prediction (Appendix F.2), and protein function prediction (Appendix F.3). Each subsection describes the application of ProteinTTT to the respective models, along with details on the data, metrics, and models. Table A3 additionally summarizes the hyperparameters used for the application of ProteinTTT to individual models.

### F.1    PROTEIN STRUCTURE PREDICTION

#### F.1.1    DATASETS

**CAMEO dataset.** To evaluate the capabilities of ProteinTTT on protein structure prediction, we employ the CAMEO validation and test sets as described in Lin et al. (2023). Specifically, the validation set was obtained by querying the CAMEO (Continuous Automated Model Evaluation) web server[4] (Robin et al., 2021) for entries between August 2021 and January 2022, while the CAMEO test set consists of entries from April 1, 2022, to June 25, 2022. Most of the entries in the CAMEO sets are predicted with high accuracy and confidence (Lin et al., 2023). Therefore, we subselect the challenging validation and test sets where customization with ProteinTTT is suitable.

Specifically, we apply two standard criteria: (1) preserving entries with ESMFold pLDDT scores below 70 to filter out high-confidence predictions (Jumper et al., 2021), and (2) selecting entries with ESM2 perplexity scores greater than or equal to 6, ensuring that the predictions are challenging due to poor sequence understanding rather than other factors. Additionally, most structures with perplexity scores below 6 are already associated with high-confidence predictions (Figure S5 in Lin et al. (2023)). After filtering, the resulting challenging validation and test sets consist of 27 (out of 378) and 18 (out of 194) targets, respectively.

---

[4]https://www.cameo3d.org/modeling

### F.1.2  METRICS

To assess the quality of the predicted protein structures with respect to the ground truth structures, we use two standard metrics averaged across the test dataset: TM-score (Zhang & Skolnick, 2004) and LDDT (Mariani et al., 2013).

**TM-score.**  The TM-score (Template Modeling score) is a metric used to assess the quality of the global 3D alignment between the predicted and target protein structures. It evaluates the structural similarity by comparing the distance between corresponding residues after superposition. The TM-score ranges from 0 to 1, where higher values indicate better alignment.

**LDDT.**  The Local Distance Difference Test (LDDT) is an alignment-free metric used to assess the accuracy of predicted protein structures. Unlike global metrics, LDDT focuses on local structural differences by measuring the deviation in distances between atom pairs in the predicted structure compared to the target structure. It is particularly useful for evaluating the accuracy of local regions, such as secondary structure elements. LDDT scores range from 0 to 100, with higher values indicating better local structural agreement.

### F.1.3  MODELS

**ESMFold.**  The ESMFold architecture comprises two key components: a protein language model, ESM2, which, given a protein sequence, generates embeddings for individual amino acids, and a folding block that, using these embeddings and the sequence, predicts the protein 3D structure along with per-amino-acid confidence scores, known as pLDDT scores. In our experiments, we use the `esmfold_v0` model from the publicly available ESMFold checkpoints[5]. Please note that we use `esmfold_v0` and not `esmfold_v1` to avoid data leakage with respect to the CAMEO test set.

**ESMFold + ProteinTTT.**  Since the ESM2 backbone of ESMFold was pre-trained in a self-supervised masked modeling regime, the application of ProteinTTT to ESMFold is straightforward. We treat ESM2 as the backbone $f$, the language modeling head predicting amino acid classes from their embeddings as the self-supervised head $g$, and the folding trunk along with the structure modules as the downstream task head $h$. After each ProteinTTT step, we run $h \circ f$ to compute the pLDDT scores, which allows us to estimate the optimal number of customization steps for each protein based on the highest pLDDT score.

Since the backbone $f$ is given by the ESM2 model containing 3 billion parameters, we apply LoRA (Hu et al., 2021) to all matrices involved in self-attention. This enables fine-tuning ESMFold + ProteinTTT on a single GPU.

**ESMFold + ME.**  Since ESMFold is a regression model, it only predicts one solution and does not have a straightforward mechanism for sampling multiple structure predictions. Nevertheless, the authors of ESMFold propose a way to sample multiple candidates (Section A.3.2 in Lin et al. (2023)). To sample more predictions, the masking prediction (ME) method randomly masks 15% (same ratio as during masked language modeling pre-training) of the amino acids before passing them to the language model. Selecting the solution with the highest pLDDT may lead to improved predicted structure. Since sampling multiple solutions with ESMFold + ME and selecting the best one via pLDDT is analogous to ESMFold + ProteinTTT, we employ the former as a baseline, running the method for the same number of steps.

**HelixFold-Single.**  HelixFold-Single is an MSA-free protein structure prediction model that combines representations from a pretrained protein language model with adapted AlphaFold2 geometric modules (EvoformerS and Structure) to directly predict atomic coordinates (Fang et al., 2023). We use the official implementation[6]

---

[5] `https://github.com/facebookresearch/esm/blob/main/esm/esmfold/v1/pretrained.py`

[6] `https://github.com/PaddlePaddle/PaddleHelix/tree/dev/apps/protein_folding/helixfold-single`

**HelixFold-Single + ProteinTTT.**  HelixFold-Single shares the main concept with ESMFold, and we combine it with ProteinTTT in the same way as in ESMFold + ProteinTTT.

**DPLM2 Bit-based.**  The DPLM2 Bit-based discrete diffusion protein language model (Hsieh et al., 2025) extends DPLM2 by using bit-wise discrete modeling to enhance structure generation capabilities (Wang et al., 2024b). DPLM2 is a multi-modal model that jointly models protein sequences and discretized structural tokens within a single discrete diffusion framework. In this work, we evaluate DPLM2 Bit-based on the task of structure prediction. Structure prediction is performed by initializing the structural tokens with masks and gradually denoising them based on the sequential tokens. We use the official implementation[7] with the standard 650M-parameter model, 100 denoising steps, and the denoising strategy set to `annealing@1.1:0.1`.

**DPLM2 Bit-based + ProteinTTT.**  To apply ProteinTTT to DPLM2 Bit-based, we use the standard masked language modeling objective (Equation (2)). See Appendix B for further discussion. Please also note that we do not use confidence function $c$ with DPLM2 Bit-based as it does not implement pLDDT or any other confidence function for protein structure prediction.

**ESM3.**  Unlike ESMFold, ESM3 is a fully multiple-track, BERT-like model (Devlin, 2018), pretrained to unmask both protein sequence and structure tokens simultaneously (along with the function tokens). The structure tokens in ESM3 are generated via a separately pre-trained VQ-VAE (Razavi et al., 2019) operating on the protein geometry. In our experiments, we use the smallest, publicly available version of the ESM3 model (`ESM3_sm_open_v0`)[8].

**ESM3 + ProteinTTT.**  We treat the Transformer encoder of ESM3 as $f$, the language modeling head decoding amino acid classes as $g$, and the VQ-VAE decoder, which maps structure tokens to the 3D protein structure, as $h$. During the customization steps, we train the model to unmask a protein sequence while keeping the structural track fully padded. During the inference, we provide the model with a protein sequence and run it to unmask the structural tokens, which are subsequently decoded with the VQ-VAE decoder. After each customization step, we run $h \circ f$ to compute the pLDDT scores, which allows us to estimate the optimal number of customization steps for each protein based on the highest pLDDT score. We choose the optimal hyperparameters by maximizing the difference in TM-score after and before applying ProteinTTT across the validation dataset.

Despite the fact that the model contains 1.4 billion parameters, even without using LoRA, ESM3 + ProteinTTT can be fine-tuned on a single NVIDIA A100 GPU. Therefore, we do not employ LoRA for fine-tuning ESM3, while this can also be possible.

**ESM3 + CoT.**  To improve the generalization and protein-specific performance of ESM3, the original ESM3 paper employs a chain of thought (CoT) procedure. The procedure unfolds in $n$ steps as follows. At each step, $1/n$ of the masked tokens with the lowest entropy after softmax on logits are unmasked. Then, the partially unmasked sequence is fed back into the model, and the process repeats until the entire sequence is unmasked. In our experiments, we set $n = 8$, which is the default value provided in the official GitHub repository.

### F.2  PROTEIN FITNESS PREDICTION

#### F.2.1  DATASETS

**ProteinGym.**  ProteinGym[9] is the standard benchmark for protein fitness prediction (Notin et al., 2024). The latest, second version of the dataset includes 217 deep mutation scanning experiments (DMSs) across different proteins. We focus on the well-established zero-shot setup of the benchmark and do not experiment with the supervised setup, as it has not yet been fully incorporated into the official codebase at the time of this study. In total, the dataset contains 2.5M mutants with annotated

---

[7] `https://github.com/bytedance/dplm`
[8] `https://github.com/evolutionaryscale/esm`
[9] `https://github.com/OATML-Markslab/ProteinGym`

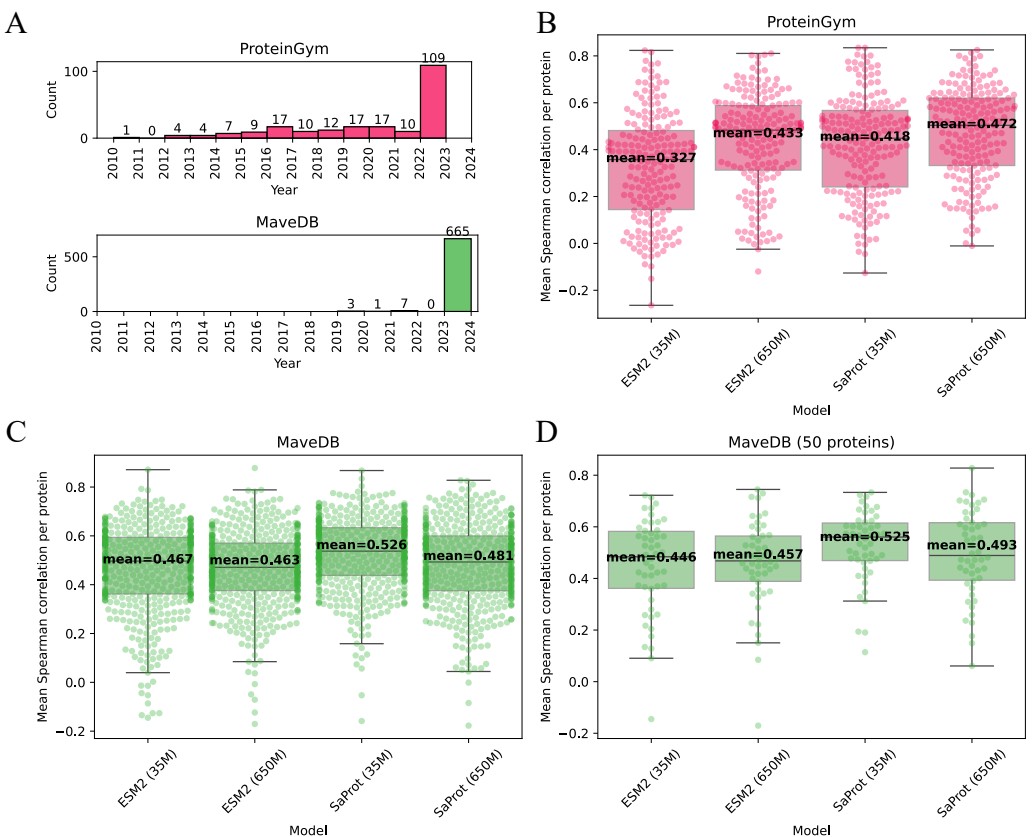

Figure A3: **Comparison of the standard ProteinGym dataset with the MaveDB dataset constructed in this work**. A) MaveDB, mined from Esposito et al. (2019), includes novel assays even after filtering to ensure distinct proteins from the comprehensive ProteinGym dataset. This is largely because most MaveDB assays post-filtering date to 2024, whereas the latest assays in ProteinGym date to 2023. B, C, D) MaveDB is of sufficient quality for model evaluation. Representative baselines, ESM2 and SaProt with both 35 million and 650 million parameters, evaluated on ProteinGym generalize effectively to MaveDB, following a similar distribution of predictions. Panel D illustrates the random subset of 50 proteins used for hyperparameter tuning for fitness prediction. Each point in the plots represents one protein and shows the Spearman correlation averaged across all assays corresponding to the protein (typically one assay per protein). The box plots standardly depict quartiles, medians, and outliers.

ground-truth fitness. Since ProteinGym does not contain a data split for the zero-shot setup, employed in this work, we use the whole dataset as the test set.

**MaveDB dataset.** To establish a validation set disjoint from ProteinGym (Notin et al., 2024), we mined MaveDB[10] (Esposito et al., 2019). As of August 1, 2024, the database contains 1178 Multiplexed Assays of Variant Effects (MAVEs), where each assay corresponds to a single protein, measuring the experimental fitness of its variants. We applied quality control filters to remove potentially noisy data. Specifically, we ensured that the UniProt identifier (Consortium, 2023) is valid and has a predicted structure available in the AlphaFold DB (Varadi et al., 2022). We also excluded assays with fewer than 100 variants, as well as those where at least one mutation had a wrongly annotated wild type or where most mutations failed during parsing. Additionally, to ensure no overlap between datasets, we removed any assays whose UniProt identifier matched with those in ProteinGym, ensuring that the validation and test sets contain different proteins.

---

[10]https://www.mavedb.org

The described methodology resulted in the MaveDB dataset comprising 676 assays (out of 1178 in the entire MaveDB) with experimental fitness annotations. This corresponds to 483 unique protein sequences and 867 thousand mutations in total. The large size of the dataset, despite the comprehensiveness of ProteinGym containing 217 assays, can be attributed to the fact that many assays in MaveDB were released after the ProteinGym construction (Figure A3A). To ensure the quality of the constructed MaveDB dataset, we validated that representative baselines from ProteinGym generalize to the new assays, following similar distributions of predictions (Figure A3B,C). Finally, for efficiently tuning hyperparameters for fitness prediction models, we sampled 50 proteins (Figure A3D), corresponding to 83 assays comprising 134 thousand variants.

### F.2.2 METRICS

Protein fitness labels are not standardized and can vary across different proteins. Nevertheless, the ranking of mutations for a single protein, as defined by fitness labels, can be used to assess the mutation scoring capabilities of machine learning models. As a result, Spearman correlation is a standard metric for evaluation.

**Spearman by phenotype.** When computing Spearman correlations, we follow the evaluation protocol proposed in ProteinGym (Notin et al., 2024). First, for each protein, we compute Spearman correlation scores between the predicted ranks of mutations and their corresponding labels. Then, we average the scores across five categories of assayed phenotypes, measuring the effects of mutations: catalytic activity ("Activity"), binding affinity to a target ("Binding"), protein expression levels in a cell ("Expression"), organism growth rate ("Organismal Fitness"), and protein thermostability ("Stability").

**Avg. Spearman.** We refer to the mean score across the five phenotype categories as "Avg. Spearman". We report the "Avg. Spearman" metric as the mean and standard deviation across five random seeds (Table 2, Table A4).

**Spearman by MSA Depth.** Following (Notin et al., 2024), we split the performance by the depth of available multiple sequence alignment (MSA), i.e., the number of homologous sequences available, as provided in ProteinGym: "Low depth", "Medium depth", and "High depth", and report the Spearman correlation for each subset individually (Table A4). Specifically, the MSA depth categories in ProteinGym are determined using the following thresholds from Hopf et al. (2017): "Low" is defined as $N_{eff}/L < 1$, "Medium" as $1 < N_{eff}/L < 100$, and "High" as $N_{eff}/L > 100$, where $N_{eff}$ represents the normalized number of effective sequences in the MSA, and $L$ is the sequence length covered in the MSA.

### F.2.3 MODELS

**ESM2.** The ESM2 model is a bidirectional, BERT-like (Devlin, 2018) Transformer trained on millions of protein sequences using masked modeling (Lin et al., 2023). The goal of protein fitness prediction is to predict the effects of mutations, and PLMs are often adapted to this task using zero-shot transfer via log odds ratio (Notin et al., 2024; Meier et al., 2021). Specifically, for a given single- or multi-point mutation, where certain amino acids $T$ are substituted from $x_i$ to $x_i^m$ for each $i \in T$, the fitness prediction via the log odds ratio is defined as:

$$\sum_{i \in T} \left( \log p(x_i^m | x_{\backslash i}) - \log p(x_i | x_{\backslash i}) \right), \tag{6}$$

where the sum iterates over mutated positions $i \in T$ with $p(x_i^m | x_{\backslash i})$ and $p(x_i | x_{\backslash i})$ denoting the predicted probabilities of the mutated amino acid and the original one (i.e., wild type), respectively. The conditionals $x_{\backslash i}$ indicate that the input sequence to the model has the position $i$ masked. In this setup, the native (unmutated) sequence, where $T = \emptyset$, has a predicted fitness of 0. Mutations with negative values represent favorable mutations, while positive values correspond to disruptive mutations. We follow the ProteinGym benchmark and use this formula (Notin et al., 2024) to evaluate the fitness prediction capabilities of ESM2. We use the implementation of ESM2 from ProteinGym.

**ESM2 + ProteinTTT.**   ESM2 can be straightforwardly customized with ProteinTTT. Specifically, we treat the Transformer encoder as the backbone $f$, and the language modeling head, which projects token embeddings to amino acid probabilities, as the pre-training head $g$. The log odds ratio given by Equation (6) serves as the task-specific head $h$, which in this case involves the pre-training head $g$ that predicts log probabilities. Overall, we apply ProteinTTT to the pre-trained ESM2 model and, after a pre-defined number of self-supervised fine-tuning steps, score mutations using Equation (6). During customization, we fine-tune all parameters in $g \circ f$ end-to-end except for token and position embeddings. When evaluating ESM2 + ProteinTTT$_{\text{MSA}}$, we use the MSAs curated by the authors of ProteinGym (Notin et al., 2024).

**SaProt.**   We also experiment with a structure-aware protein language model, SaProt (Su et al., 2023). SaProt builds off the ESM2 model but incorporates structural information from predicted protein structures. Specifically, SaProt uses the same Transformer architecture but expands its vocabulary by combining the 20 standard amino acid tokens with 20 structural tokens from the 3Di vocabulary, increasing the total alphabet size to 400. The 3Di tokens capture the geometry of the protein backbone and are generated using VQ-VAE (Razavi et al., 2019), which projects continuous geometric information into discrete tokens and was trained as part of the Foldseek method (van Kempen et al., 2022).

Since SaProt is also a protein language model, it also uses Equation (6) to score variants. However, please note that SaProt, as implemented in ProteinGym (Notin et al., 2024), uses a slightly different version of the log odds ratio. In SaProt, the conditions in the log probabilities in Equation (6) are replaced with $x_{\setminus T}$ instead of $x_{\setminus i}$, not assuming the independence of substitutions. During customization with ProteinTTT, we only mask sequential information and leave the structural part of the tokens unchanged, reflecting the original pre-training setup. We use the implementation of SaProt from ProteinGym[9].

**SaProt + ProteinTTT.**   Since the architecture of SaProt is based on ESM2, the ProteinTTT components $f$, $g$, and $h$ remain the same. It means that customization can be applied to the model in the same way as in the case of ESM2 + ProteinTTT discussed above.

**ProSST.**   We experiment with the state-of-the-art fitness predictor, ProSST (Li et al., 2024). ProSST primarily improves upon SaProt (Su et al., 2023) by incorporating a larger vocabulary of structural tokens and employing disentangled attention mechanisms. Instead of relying on the 3Di alphabet optimized for protein structure search with Foldseek (van Kempen et al., 2022), Li et al. (2024) pre-train a new autoencoder to denoise corrupted protein backbones and cluster the resulting latent space using the $K$-means algorithm (Lloyd, 1982). Notably, optimal performance for fitness prediction is achieved with $K = 2048$ tokens, compared to just 20 in the 3Di vocabulary used by SaProt. We adopt this model in our experiments. Additionally, disentangled attention in ProSST enhances information propagation between sequence and structure within its Transformer blocks, further improving prediction performance. The model has 110M parameters in total.

ProSST, similarly to ESM2 and SaProt, is pre-trained using masked language modeling applied to protein sequence tokens. To score mutations on the ProteinGym benchmark (Notin et al., 2024), ProSST also uses the log-odds ratio, but in a slightly different way compared to ESM2 and SaProt. Specifically, ProSST performs a single forward pass to predict log probabilities, which are then used to score all mutations. Formally, this approach modifies the log probability condition in Equation (6), replacing $x_{\setminus i}$ with $x$.

**ProSST + ProteinTTT.**   Similarly to ESM2 and SaProt, we treat the Transformer encoder in ProSST as the backbone $f$, the masked language modeling head as the pre-training head $g$, and the log-odds ratio formula as the task-specific head $h$.

**ProGen2.**   For fitness prediction, we additionally experiment with one of the major autoregressive protein language models, ProGen2 (Nijkamp et al., 2023). Specifically, we experiment with ProGen of two sizes: ProGen2-small (151M parameters) and ProGen2-large (2.7B parameters). We obtain the pre-trained weights from the official GitHub repository[11]. For ProGen2-large inference, we use floating-point 16 precision for computational efficiency.

---

[11]https://github.com/salesforce/progen/tree/main/progen2

**ProGen2 + ProteinTTT.**   To demonstrate the applicability of ProteinTTT in an autoregressive setting, we apply it to the ProGen2 (Nijkamp et al., 2023) language model. To perform the customization, we use the standard next-token prediction objective on a single target protein, following Equation (4). Please see Appendix B for details.

**MSA Transformer.**   Finally, we experiment with MSA Transformer for fitness prediction (Rao et al., 2021). Similar to ESM2 (Lin et al., 2023), MSA Transformer is pre-trained on large protein sequence datasets; however, it is trained on multiple sequence alignments (MSAs) rather than individual sequences.

Since MSA Transformer is also a protein language model, it can be used for fitness prediction in the same way as ESM2, as discussed above, by computing the log-odds ratio over the first sequence in the MSA in this case. We reproduce the results of MSA Transformer on the ProteinGym benchmark with two modifications: (1) we sample a weighted subset of 32 sequences from each MSA instead of 400, and (2) we use only one random seed instead of five for ensembling. These changes significantly reduce computational time while also slightly improving performance compared to the results reported in ProteinGym. This improvement may be explained by the fact that the performance of MSA Transformer saturates with increasing MSA input size (Figure 4 in Rao et al. (2021)).

**MSA Transformer + ProteinTTT.**   We experiment with customizing MSA Transformer to MSA subsamples of varying sizes, ranging from a single target sequence (i.e., customization via Equation (2) with ProteinTTT) to the full MSA subset of 32 sequences (i.e., customization via Equation (5) with ProteinTTT$_{MSA}$). We observe that applying ProteinTTT$_{MSA}$ to MSA Transformer with a batch size of 32 disrupts performance, while reducing the input MSA subsample size mitigates this effect. Ultimately, MSA Transformer + ProteinTTT results in a slight performance improvement.

### F.3    PROTEIN FUNCTION PREDICTION

#### F.3.1    DATASETS

**TPS dataset.**   For the evaluation of terpene substrate classification, we use the largest available dataset of characterized TPS enzymes from Samusevich et al. (2025) and repurpose the original 5-fold cross-validation schema. We focus on the most challenging TPS sequences, defined as those predicted by the TPS detector, proposed by the dataset authors, with confidence scores below 0.8. This filtering results in 104, 98, 113, 100, 97 examples in the individual folds.

**setHard.**   For the test evaluation of subcellular location prediction, we use the setHard dataset constructed by Stärk et al. (2021). The dataset was redundancy-reduced, both within itself and relative to all proteins in DeepLoc (Almagro Armenteros et al. (2017); next paragraph), a standard dataset used for training and validating machine learning models. The setHard dataset contains 490 protein sequences, each annotated with one of ten subcellular location classes, such as "Cytoplasm" or "Nucleus". Since we use ESM-1b (Rives et al., 2021) in our experiments with the dataset, we further filter the data to 432 sequences that do not exceed a length of 1022 amino acids. This step, consistent with Stärk et al. (2021), ensures that ESM-1b can generate embeddings for all proteins.

**DeepLoc.**   For hyperparameter tuning in the subcellular location prediction task, we use the test set from the DeepLoc dataset (Almagro Armenteros et al., 2017). Similar to setHard, DeepLoc assigns labels from one of ten subcellular location classes. The dataset contains 2768 proteins, which we further filter to 2457 sequences that do not exceed a length of 1022 amino acids, ensuring compatibility with the embedding capabilities of ESM-1b. Since setHard was constructed to be independent of DeepLoc, setHard provides a leakage-free source of data for validation.

#### F.3.2    METRICS

**mAP, AUROC.**   The TPS substrate prediction problem is a 12-class multi-label classification task over possible TPS substrates. Therefore, we assess the quality of the predictions using standard multi-label classification metrics such as mean average precision (mAP) and area under the receiver operating characteristic curve (AUROC) averaged across individual classes. These metrics were used in the original EnzymeExplorer paper (Samusevich et al., 2025). We report the performance by

averaging the metric values concatenated across all validation folds from the 5-fold cross-validation schema.

**Accuracy, MCC, F1-score.** To evaluate the performance of subcellular location prediction methods, we use standard classification metrics as employed in Stärk et al. (2021). Accuracy standardly measures the ratio of correctly classified proteins, while Matthew's correlation coefficient for multiple classes (MCC) serves as an alternative to the Pearson correlation coefficient for classification tasks (Gorodkin, 2004). The F1-score, the harmonic mean of precision and recall, evaluates performance from a retrieval perspective, balancing the trade-off between false positives and false negatives.

### F.3.3 Models

**EnzymeExplorer.** EnzymeExplorer is a state-of-the-art method for the classification of terpene synthase (TPS) substrates (Samusevich et al., 2025). The model consists of two parallel tracks. Given a protein sequence, EnzymeExplorer first computes its ESM-1v embedding (Meier et al., 2021) and a vector of similarities to the functional domains of proteins from the training dataset, based on unsupervised domain segmentation of AlphaFold2-predicted structures (Jumper et al., 2021). The ESM-1v embedding and the similarity vector are then concatenated and processed by a separately trained random forest, which predicts TPS substrate class probabilities.

In our experiments, we use the "PLM only" version of the model, which leverages only ESM-1v embeddings. This version exhibits a minor performance decrease compared to the full model but exactly follows a Y-shaped architecture, allowing us to validate the effectiveness of ProteinTTT for predicting TPS substrates. We use the implementation of EnzymeExplorer available at the official GitHub page [12].

**EnzymeExplorer + ProteinTTT.** When applying ProteinTTT to EnzymeExplorer, we treat the frozen ESM-1v model as a backbone $f$, its language modeling head as a self-supervised head $g$, and the random forest classifying TPS substrates as a downstream supervised head $h$.

**Light Attention.** We use Light attention (Stärk et al., 2021) as a representative baseline for subcellular location prediction. Light attention leverages protein embeddings from a language model, which in our case is ESM-1b (Rives et al., 2021). The model processes per-residue embeddings via a softmax-weighted aggregation mechanism, referred to as light attention, which operates with linear complexity relative to sequence length and enables richer aggregation of per-residue information, as opposed to standard mean pooling. We re-train the model using ESM-1b embeddings on the DeepLoc dataset (Almagro Armenteros et al., 2017) using the code from the official GitHub page [13].

**Light attention + ProteinTTT.** When applying ProteinTTT to Light attention, we treat the frozen ESM-1b as the backbone $f$, the language modeling head of ESM-1b as the self-supervised head $g$, and the Light attention block as the fine-tuning head $h$.

## G  Case study details

### G.1  Modeling antibody-antigen loops

We download the SAbDab dataset from the official website[14](Dunbar et al., 2014). We apply ProteinTTT to targets with low-confidence ESMFold predictions (pLDDT < 70) and remove sequences longer than 400 residues due to GPU memory limitations. This results in a final set of 175 antibody and 814 antigen chains. We predict the full structures using ESMFold+ProteinTTT (with the same hyperparameters tuned on the CAMEO validation set specified in Table A3) and compute LDDT scores against the corresponding PDB structures to assess local errors, which are particularly relevant for loop regions. For antibodies, we evaluate the complete structures, while for complementarity-determining regions (CDRs), we extract the CDR substructures as annotated in SAbDab according to Chothia numbering (Chothia & Lesk, 1987) and calculate LDDT on these regions.

---

[12]https://github.com/pluskal-lab/EnzymeExplorer
[13]https://github.com/HannesStark/protein-localization
[14]https://opig.stats.ox.ac.uk/webapps/sabdab-sabpred/sabdab

Table A3: **Hyperparameters used for adapting ProteinTTT to individual models.** The optimal hyperparameters were estimated using validation datasets corresponding to each of the considered tasks: *Fitness prediction*, *Structure prediction*, and *Function prediction*. Comma-separated lists show the values used for hyperparameter grid search, while the final values selected for computing the test results are highlighted in **bold**. Low-rank adaptation (LoRA) was only used with ESMFold, containing 3 billion parameters in the ESM2 backbone. Please note that we did not tune the number of customization steps, as adjusting the learning rate and batch size effectively controls the expected performance under the fixed number of steps, as shown in Figure A10. Therefore, we used 30 steps in all our experiments. The only exception was ESM3 + ProteinTTT, where the number of steps was set to 50 during initial experiments with different models/tasks conducted in parallel before standardizing the number of steps to 30. Bidirectional methods marked with an asterisk ("*") used a slightly different calculation of the loss function. Specifically, the loss was propagated over all tokens, including special and non-masking tokens, while averaging the loss across all tokens simultaneously, rather than first averaging over sequences. This approach was used in the early stages of development, and we provide it in our codebase via `loss_kind = "unnormalized_cross_entropy"`. Please note that MSA Transformer always uses 1 MSA in a batch and the "Batch size" represents the number of sequences in this MSA with the target sequence always present as the first one.

| | Learning rate | Batch size | Grad. acc. steps | Steps (Conf. func. $c$) | LoRA rank $r$ | LoRa $\alpha$ |
|---|---|---|---|---|---|---|
| *Fitness prediction* | | | | | | |
| ESM2 (35M) + ProteinTTT * | 4e-5, **4e-4**, 4e-3 | **4** | 4, 8, **16**, 32, 64 | **30** | - | - |
| ESM2 (650M) + ProteinTTT * | **4e-5**, 4e-4, 4e-3 | **4** | 4, 8, **16**, 32 | **30** | - | - |
| ProGen2-small (151M) + ProteinTTT | 4e-5, **4e-4**, 4e-3 | **4** | **4** | 1, 5, 10, **15**, 20 | - | - |
| ProGen2-large (2.7B) + ProteinTTT | **1e-5** | **4** | **4** | **10**, 15, 20 | **4**, 8 | 8, **16** |
| SaProt (35M) + ProteinTTT * | 4e-5, **4e-4**, 4e-3 | **4** | 4, **8**, 16, 32 | **30** | - | - |
| SaProt (650M) + ProteinTTT * | **4e-5**, 4e-4, 4e-3 | 2, **4** | 4, 8, **16**, 32 | **30** | - | - |
| ProSST (K=2048) + ProteinTTT * | **1e-5**, 4e-5, 4e-4, 4e-3 | **4** | 4, **8**, 16, 32 | **30** | - | - |
| ESM2 (650M) + ProteinTTT$_{MSA}$ * | 4e-6, 1e-5, 4e-5, 4e-4, **4e-3** | **4** | **2**, 4 | 50, **100** | - | - |
| MSA Transformer + ProteinTTT | 1e-6, 3e-6, 1e-5, 3e-5, **1e-4** | **1**, 4, 8, 16, 32 | 1, 2, 4, **8** | **30** | - | - |
| *Structure prediction* | | | | | | |
| ESM3 + ProteinTTT | 1e-4, 4e-4, **1e-3** | **2** | **1**, 4, 16 | **50 (pLDDT)** | - | - |
| DPLM2 Bit-based + ProteinTTT | 4e-6, 4e-5, **4e-4**, 4e-3 | 2, 4, **8** | **2**, 4, 8 | **10** | - | - |
| HelixFold-Single + ProteinTTT | **4e-4**, 1e-3 | **4**, 8, 16 | **1** | **30 (pLDDT)** | - | - |
| ESMFold + ProteinTTT | **4e-4** | **4** | **4**, 8, 32, 64 | **30 (pLDDT)** | 4, **8**, 32 | 8, 16, **32** |
| *Function prediction* | | | | | | |
| EnzymeExplorer + ProteinTTT | **4e-4**, 1e-3 | **2** | **2**, 4, 8 | **30** | - | - |
| Light attention + ProteinTTT | 4e-4, 1e-3, **3e-3** | **2** | **2**, 4 | **30** | - | - |

## G.2 Expanding known structures of viral proteins

We use BFVD version `archived/2023_02_v2`[15]. This version contains maximum-pLDDT structures from predictions generated by two strategies: (i) ColabFold (Mirdita et al., 2022) with MSAs constructed using Logan (Chikhi et al., 2024), and (ii) ColabFold with 12 additional recycle steps and MSAs constructed using Logan. In Figure 5, we also report pLDDT values for BFVD version `archived/2023_02_v1`, where structures are simply obtained from ColabFold with MSAs from Logan, i.e., strategy (i). We re-predict structures using ESMFold and ESMFold+ProteinTTT for sequences with length < 450 due to GPU memory constraints. We use the same hyperparameters tuned on the CAMEO validation set, as specified in Table A3, with the exception of 20 instead of 30 steps for computational efficiency.

## H Extended results

In this section, we provide additional results on test sets (Appendix H.1), discuss validation performance (Appendix H.2), and analyze the runtime performance of customization (Appendix H.3).

### H.1 Detailed test performance

In this section, we provide details on the test performance. Specifically, Table A4 shows that customization with ProteinTTT primarily enhances performance on challenging targets, characterized by a low number of similar proteins in sequence databases, as measured by MSA depth. Additionally, we provide a qualitative example illustrating how ProteinTTT substantially improves the correlation

---

[15]`https://bfvd.steineggerlab.workers.dev`

between ESM2-predicted fitness and ground-truth stability by better identifying disruptive mutations in the protein core (Figure A5).

Next, Figure A6 shows the distribution of ProteinTTT effects: in many cases, customization has minimal impact on performance; often, it leads to substantial improvements; and in rare cases, customization results in a decrease in performance. This positions ProteinTTT as a method for enhancing prediction accuracy, while a comprehensive analysis of its failure modes remains an important direction for future research. While we demonstrate these effects using a protein folding example, we observe a similar distribution of ProteinTTT impact across the tasks.

We also observe that the overall trend of customization with ProteinTTT generally leads to improved performance, with robust consistency across random seeds. However, the progression of the performance curve can be rugged, particularly in classification tasks, where substantial changes in the underlying representations are required to shift the top-predicted class in the discrete probability distribution (Figure A12).

## H.2 VALIDATION PERFORMANCE

This section discusses the performance of ProteinTTT on validation data. Table A5 illustrates the validation performance of the tested methods for fitness prediction on our newly constructed MaveDB dataset. ProteinTTT enhances the performance of all the methods.

Next, we discuss the hyperparameter optimization. Table A3 provides the grid of hyperparameters explored for each model and its size, as well as specifies the optimal hyperparameters suitable for downstream applications. Figure A10 demonstrates the trend of hyperparameter tuning with optimal hyperparameter combination balancing underfitting and overfitting to a single target protein. While most of reasonable hyperparameter configurations lead to overall improvements when using customization with ProteinTTT, poorly chosen hyperparameters can have detrimental effects due to rapid overfitting. However, with a reliable predicted confidence measure, such as pLDDT, the appropriate customization step for each protein can be selected to mitigate overfitting. Figure A11 demonstrates that when using ESM3 + ProteinTTT with pLDDT-based step selection for protein structure prediction, all hyperparameter configurations result in improved performance compared to the base ESM3 model.

## H.3 RUNTIME PERFORMANCE

In this section, we demonstrate that customization with ProteinTTT can be done efficiently, with an acceptable computational overhead. Specifically, we show that ESMFold, known for being a faster alternative to more performant methods such as AlphaFold2 (Jumper et al., 2021) or AlphaFold3 (Abramson et al., 2024), still remains in the category of lightweight methods even with ProteinTTT customization (Figure A4).

This observation highlights the practical utility of ProteinTTT. For example, ESMFold enabled structural characterization of large metagenomics data (>617 million metagenomic sequences), which would be infeasible with AlphaFold2 (Lin et al., 2023). Nevertheless, the original ESMFold has high confidence predictions only for 36% of sequences from the metagenomic database, while the other 392 million sequences remain with low or medium confidence predictions. At the same time, ESMFold + ProteinTTT enables more accurate predictions compared to the original ESMFold (Figure A6 suggests that ESMFold + ProteinTTT significantly improves predictions in almost 40% of challenging sequences). It means that applying ESMFold + ProteinTTT to these remaining sequences could significantly expand the metagenomic atlas characterized by ESMFold. Here, we illustrate this on a similar case study by applying ESMFold + ProteinTTT to more than 300 thousand viral proteins in BFVD (Section 5.2)

## I  LIMITATIONS AND FUTURE WORK

We see two main limitations of the current version of ProteinTTT, which we discuss in detail below.

**Extension to other model types and tasks.** The current form of the method is only applicable to protein language models (PLMs), i.e., Transformer-based (Vaswani, 2017) models pre-trained

using bidirectional masked language modeling (Rives et al., 2021) or autoregressive next-token prediction (Nijkamp et al., 2023). Nevertheless, the concept of test-time training can also be extended to many other models in computational biology, which presents exciting opportunities for future research, as our work demonstrates the high potential of this paradigm for the field of computational biology. For instance, our central experiments in Section 4.1 use ESMFold (Lin et al., 2023), which is known to often underperform (Lin et al., 2023) more specizalized multiple sequence alignment (MSA)-based structure predictors such as AlphaFold2 (Jumper et al., 2021), AlphaFold-Multimer (Evans et al., 2021), AlphaFold3 (Abramson et al., 2024), or Boltz-2 (Passaro et al., 2025).

Nevertheless, all of these models can also be extended with test-time training akin to ProteinTTT. AlphaFold2, and subsequently AlphaFold-Multimer, use masked modeling of MSA as one of the training objectives to learn powerful pairwise representations in Evoformer. The Evoformer backbone could therefore be updated at test time to obtain more powerful representation of one input MSA at a time using the ProteinTTT objective (Section 3.1). While AlphaFold3 and Boltz-2 do not use masked modeling, they can still be customized in a self-supervised way, for example using an optimization through distogram (Cho et al., 2025). Implementing the variants of ProteinTTT for the aforementioned models could enable customized structure prediction of protein multimers and protein-ligand complexes.

Beyond structure prediction, test-time customization could also benefit *de novo* protein design. Our results with autoregressive ProGen2 on fitness prediction suggest that ProteinTTT can improve sequence design (Table 2). Similarly, although our experiments with ESM3 are currently conducted in the context of structure prediction (Table 1), ProteinTTT can be straightforwardly applied to ESM3 for protein design tasks such as inverse folding or structure inpainting by applying ProteinTTT to the corresponding ESM3 input tracks. Furthermore, BoltzGen (Stark et al., 2025) can be extended with test-time customization in a manner analogous to Boltz-2, discussed above, due to their shared architecture. Performing ProteinMPNN (Dauparas et al., 2022) customization on part of a protein to guide generation of the remaining structure or its binder, as well as customizing RFdiffusion (Watson et al., 2023) to a target structure for binder design, represent promising opportunities in protein design with the potential for higher success rates.

**Better control over failure cases.** The failure modes of ProteinTTT are not yet fully understood. For instance, combining ESMFold with ProteinTTT decreases performance for several proteins in the CAMEO test set (Figure A6). A detailed analysis of these cases shows that the degradation can be attributed to ambiguity in the evaluation itself (Figure A13). These examples illustrate the challenge of identifying a general reason for the occasional degradation of performance. As discussed in Appendix H.2, confidence functions (such as pLDDT in structure prediction) allow effectively eliminating overfitting to a single protein and thereby mitigating such failure cases, making confidence prediction an essential component of customization.

While confidence functions begin to emerge across tasks, such as fitness prediction (Gurev et al., 2025; Nijkamp et al., 2023) and inverse folding (Shuai et al., 2025), they are not yet universally available for use with ProteinTTT. In particular, for fitness (Section 4.2) and function (Section 4.3) prediction, controlling failure cases remains more challenging due to the absence of a reliable confidence metric. This motivates the development of general, task-agnostic, unsupervised confidence measures (for example, perplexity-based estimates (Gurev et al., 2025)) or a dedicated confidence prediction module within ProteinTTT (Abramson et al., 2024; Jumper et al., 2021). Another promising direction is deriving confidence estimates from mechanistic interpretability of protein language models (Hübotter et al., 2025; Simon & Zou, 2025; Zhang et al., 2024).

Table A4: **ProteinTTT performance on ProteinGym depending on MSA depth.** MSA depth reflects the number of available proteins similar to the target protein and, when using large protein language models, can be interpreted as a measure of the representation of similar proteins in the training data (Appendix F.2.2). Customization with ProteinTTT primarily improves performance on difficult targets, with low MSA depth. Standard deviations are calculated over 5 random seeds but are omitted in the right panel for brevity, where the maximum standard deviation does not exceed 0.0004.

| | Avg. Spearman ↑ | Spearman by MSA depth ↑ | | |
| --- | --- | --- | --- | --- |
| | | Low depth | Medium depth | High depth |
| ESM2 (35M) (Lin et al., 2023) | 0.3211 | 0.2394 | 0.2707 | 0.451 |
| ESM2 (35M) + ProteinTTT (Ours) | **0.3407 ± 0.00014** | **0.2445** | **0.3144** | **0.4598** |
| ProGen2-small (151M) (Nijkamp et al., 2023) | 0.3255 | 0.2974 | 0.3136 | 0.3765 |
| ProGen2-small (151M) + ProteinTTT (Ours) | **0.3591** ± 0.0002 | **0.3319** | **0.3636** | **0.3917** |
| ProGen2-large (2.7B) (Nijkamp et al., 2023) | 0.3724 | 0.3504 | **0.3808** | 0.4090 |
| ProGen2-large (2.7B) + ProteinTTT (Ours) | **0.3817** ± 0.00158 | **0.3627** | 0.3791 | **0.4152** |
| SaProt (35M) (Su et al., 2023) | 0.4062 | 0.3234 | 0.3921 | 0.5057 |
| SaProt (35M) + ProteinTTT (Ours) | **0.4106 ± 0.00004** | **0.3253** | **0.3972** | **0.5091** |
| ESM2 (650M) (Lin et al., 2023) | 0.4139 | 0.3346 | 0.4063 | **0.5153** |
| ESM2 (650M) + ProteinTTT (Ours) | **0.4153 ± 0.00003** | **0.3363** | **0.4126** | 0.5075 |
| SaProt (650M) (Su et al., 2023) | 0.4569 | 0.3947 | **0.4502** | **0.5448** |
| SaProt (650M) + ProteinTTT (Ours) | **0.4583 ± 0.00001** | **0.3954** | 0.4501 | 0.5439 |
| ProSST (K=2048) (Li et al., 2024) | 0.5068 | 0.4731 | **0.5107** | 0.5749 |
| ProSST (K=2048) + ProteinTTT (Ours) | **0.5087 ± 0.00004** | **0.4809** | 0.5104 | **0.5750** |

Table A5: **Performance of ProteinTTT on the MaveDB dataset.** In this work, we use our newly constructed MaveDB dataset as a validation fold for tuning the ProteinTTT hyper-parameters for fitness prediction. For computational efficiency, we only select a subset of 50 proteins (Appendix F.2.1) and do not run customization across multiple random seeds to estimate standard deviations. The performance shown was calculated by first aggregating correlations per assay, and then per protein (some assays correspond to the same protein).

| | Avg. Spearman ↑ |
| --- | --- |
| ESM2 (35M) (Lin et al., 2023) | 0.4458 |
| ESM2 (35M) + ProteinTTT (Ours) | **0.4593** |
| ESM2 (650M) (Lin et al., 2023) | 0.4568 |
| ESM2 (650M) + ProteinTTT (Ours) | **0.4604** |
| SaProt (650M) (Su et al., 2023) | 0.4926 |
| SaProt (650M) + ProteinTTT (Ours) | **0.4926** |
| SaProt (35M) (Su et al., 2023) | 0.5251 |
| SaProt (35M) + ProteinTTT (Ours) | **0.5271** |
| ProSST (K=2048) (Li et al., 2024) | 0.5444 |
| ProSST (K=2048) + ProteinTTT (Ours) | **0.5462** |

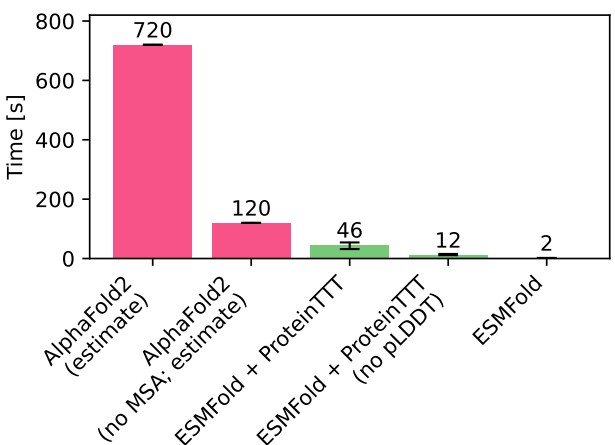

Figure A4: **Running time of ESMFold + ProteinTTT.** For ESMFold and its variants, the median and interquartile ranges of running times on the CAMEO test set are shown using a single NVIDIA A100 GPU. For AlphaFold2, we use estimates from Lin et al. (2023). Specifically, a forward pass through AlphaFold2 is approximately 60 times more computationally expensive than ESMFold (e.g., `AlphaFold2 (no MSA; estimate)`: $2 \times 60 = 120$ seconds), with additional MSA construction taking at least 10 minutes using standard pipelines (`AlphaFold2 (estimate)`: $2 \times 60 + 10 \times 60 = 720$ seconds). ESMFold + ProteinTTT (30 steps) involves LoRA parameter updates, along with forward passes at each customization step to estimate pLDDT and select the structure with the highest predicted confidence. Disabling pLDDT significantly reduces computational overhead (`ESMFold + ProteinTTT (no pLDDT)` compared to `ESMFold + ProteinTTT`), but may require careful parameter tuning (Appendix H.2). Overall, ESMFold + ProteinTTT maintains the speed advantage of ESMFold, and is at least an order of magnitude faster than AlphaFold2.

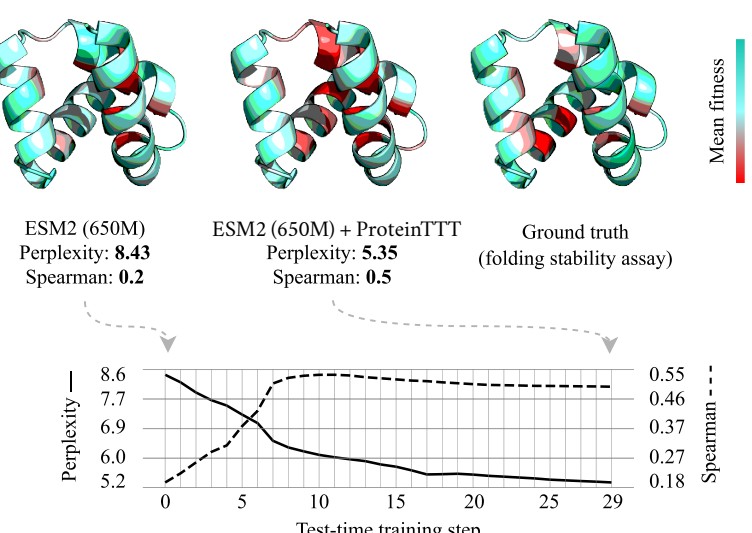

Figure A5: **Example of protein fitness prediction upon single-sequence model customization with ProteinTTT.** Fitness predictions from ESM2 (650M) show poor correlation with experimental fitness values in the ProteinGym test set measured by the stability assay "UBR5_HUMAN_Tsuboyama_2023_1I2T" (Tsuboyama et al., 2023) (left). ESM2 + ProteinTTT achieves significantly higher correlation, likely due to improved detection of disruptive mutations in the protein core that impact protein stability (middle). The ground-truth fitness data aligns with the customized model, showing that residues crucial for stability (i.e., having negative mean fitness) are concentrated in the protein core (right). Residue colors represent the mean fitness upon all single-point substitutions (with the exception of several missing mutations in the ground-truth data), with red indicating residues where mutations have detrimental effects on average.

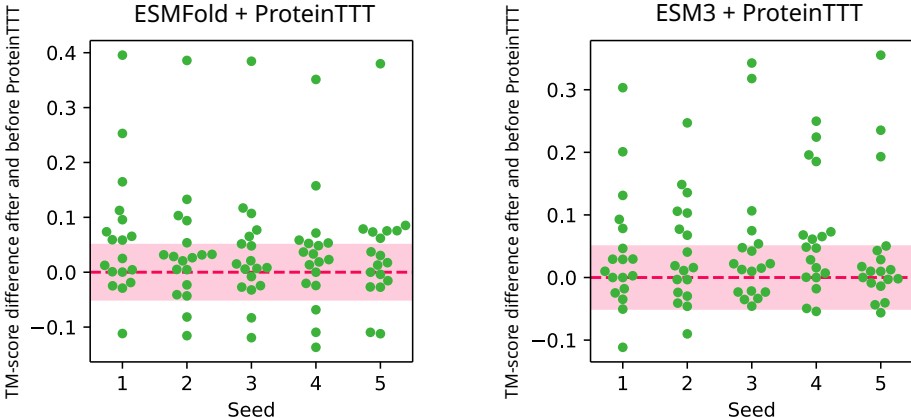

Figure A6: **Per-protein performance of ESMFold + ProteinTTT and ESM3 + ProteinTTT on the CAMEO test set.** The y-axis shows the change in TM-score after applying customization with ProteinTTT, with higher values indicating improvement. The x-axis represents performance across five random seeds. The red dashed line marks no change in TM-score (TM-score difference = 0), and the pink band represents minor changes in TM-score ($-0.05 <$ TM-score difference $< 0.05$), which we do not consider significant. Each point in the swarm plot corresponds to a single protein from the CAMEO test set. On average, applying ProteinTTT to ESMFold improves the structure predictions for 7 out of 18 proteins, with 2 showing degradation. The rest of the proteins are not significantly affected. Similarly, applying ProteinTTT to ESM3 results in 6 improvements out of 18 proteins, with 1 case of degradation.

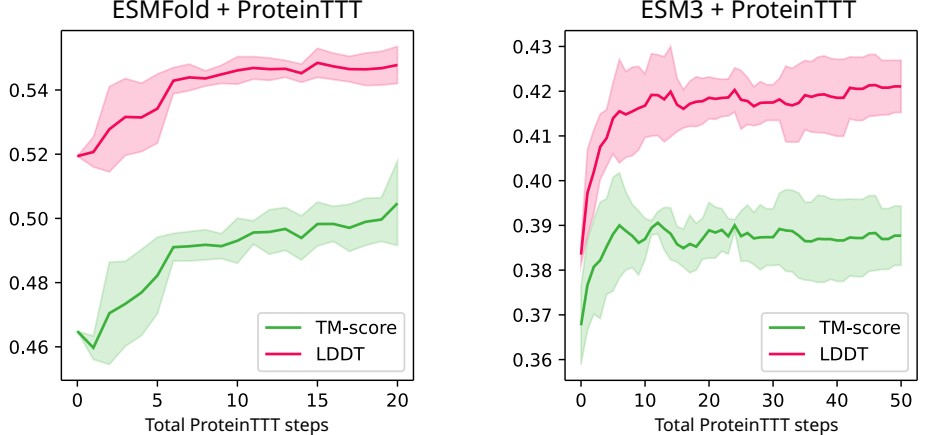

Figure A7: **Test performance of ESMFold + ProteinTTT and ESM3 + ProteinTTT on the CAMEO test set depending on the total number of customization steps.** The x-axis shows the averaged performance across all test proteins, with error bars representing the standard deviation across five random seeds. The y-axis metrics correspond to the structure with the highest pLDDT score up to the given step. While an increased number of ProteinTTT steps generally enhances performance, only a few steps (e.g., five) may suffice to achieve significant performance improvement.

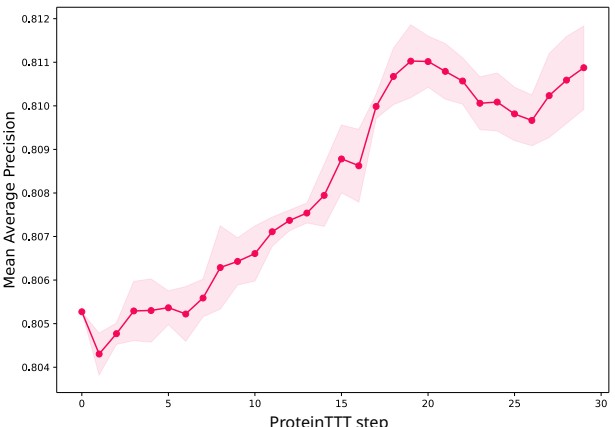

Figure A8: **Test performance of EnzymeExplorer + ProteinTTT across customization steps.** The performance is averaged across all 512 proteins in the dataset, with error bars representing the standard deviation across 5 random seeds.

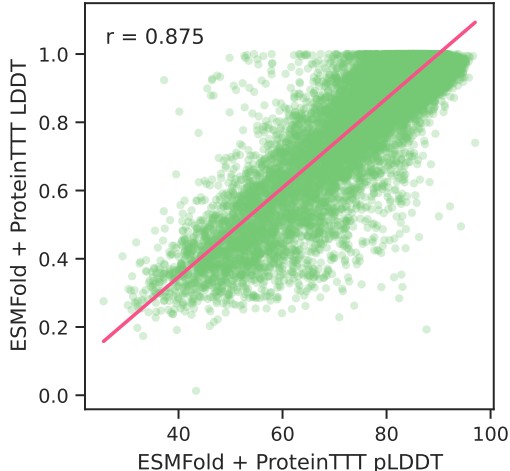

Figure A9: **ESMFold + ProteinTTT pLDDT correlates with ESMFold + ProteinTTT LDDT.** The evaluation was performed on 17,582 AlphaFold2 reference structures from the BFVD database with pLDDT > 90. Here, $r = 0.875$ denotes the Pearson correlation coefficient.

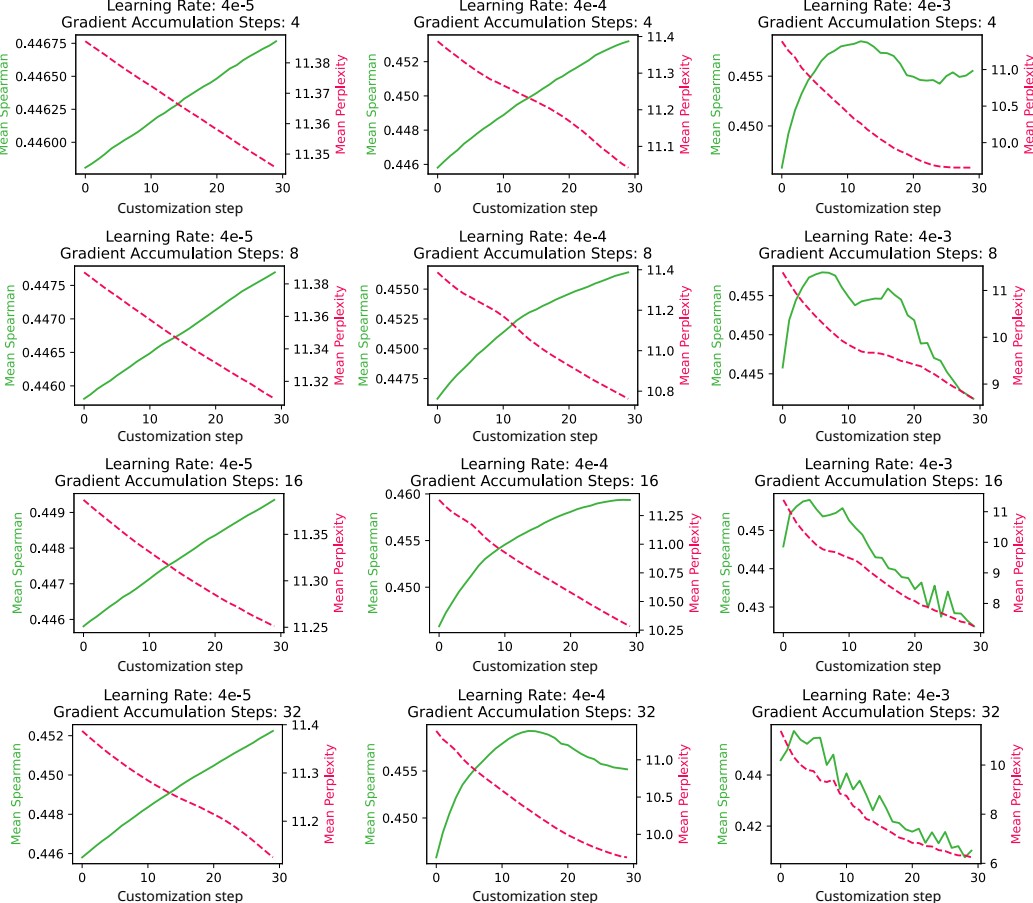

Figure A10: **Dependence on ProteinTTT hyperparameters for customized fitness prediction.** Each plot shows the progression of Spearman correlation (green) increasing alongside a decrease in perplexity (pink) for each customization step, averaged across all assays in the MaveDB validation dataset. The model used is ESM2 (35M) + ProteinTTT, and the grid displays the combinations of different numbers of gradient accumulation steps (i.e., effective batch sizes; shown in rows, increasing from top to bottom) and learning rates (columns, increasing from left to right). As the learning rate increases and the number of gradient accumulation steps grows, the model reaches peak performance more quickly but begins to overfit to a target protein. The optimal hyperparameter combination (learning rate = 4e-4, gradient accumulation steps = 16) lies near the center of the grid, balancing between underfitting and overfitting to a target protein. Notably, the figure demonstrates that, although ProteinTTT involves three main hyperparameters (batch size, learning rate, and the number of steps), there are effectively only two degrees of freedom controlling the performance of the model. In other words, by keeping the number of steps constant (e.g., 30), the expected performance can be controlled by adjusting the learning rate and the batch size.

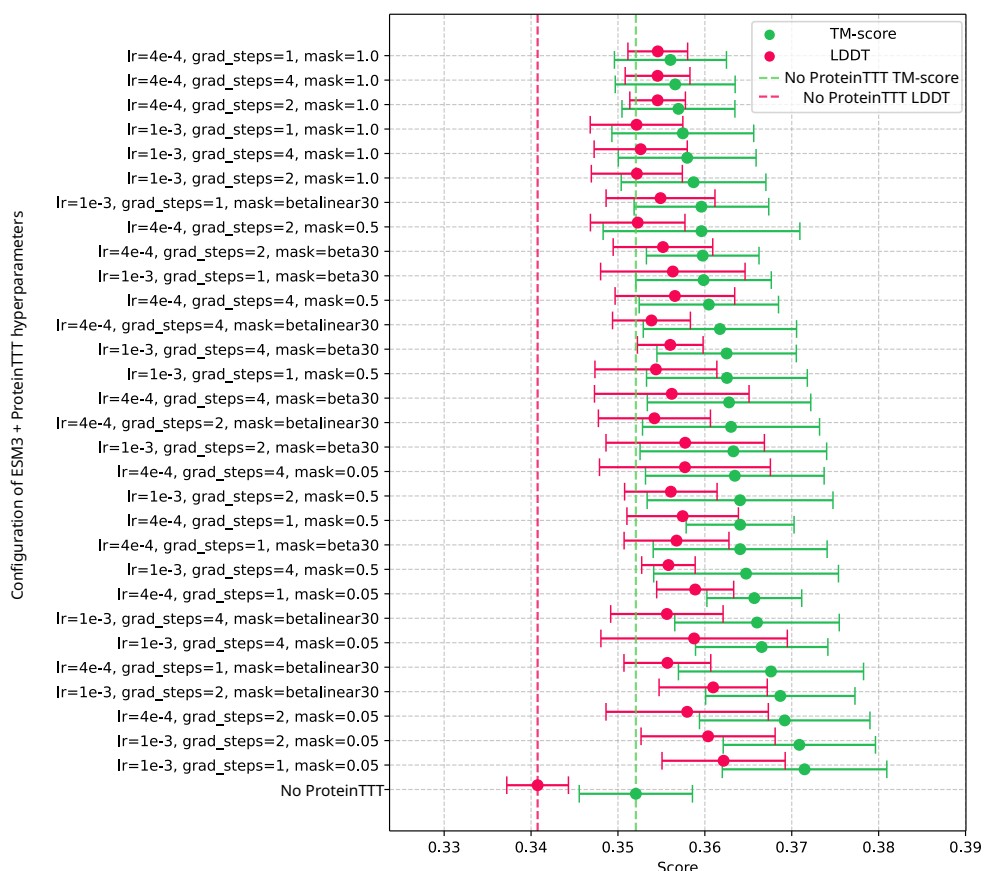

Figure A11: **Hyperparameter search for protein structure prediction with ESM3 + ProteinTTT.** We conducted a comprehensive grid search based on three key hyperparameters: learning rate (denoted as "lr"), number of gradient accumulation steps (denoted as "grad_steps"; with the batch size of two), and masking strategy (denoted as "mask"). We explored two learning rates, 4e-4 and 1e-3, three gradient accumulation step values of 1, 4, and 16, and five different masking strategies: uniform sampling of 0.05, 0.5, and 1.0 fractions of amino acids, as well as the "beta30" and "betalinear30" distributions proposed in the ESM3 paper (Hayes et al., 2024). Each row in the table presents the mean TM-score and LDDT metrics with standard deviations across five random seeds on the CAMEO validation fold. The last row, denoted as "No ProteinTTT", shows the performance of ESM3 without customization. The results indicate that ESM3 + ProteinTTT is robust to the choice of hyperparameters and consistently outperforms the base model across all configurations. We selected the configuration from the last row (excluding "No ProteinTTT") to compute the results on the test fold. For the hyperparameter search, we used 30 customization steps instead of 50 to reduce computation time.

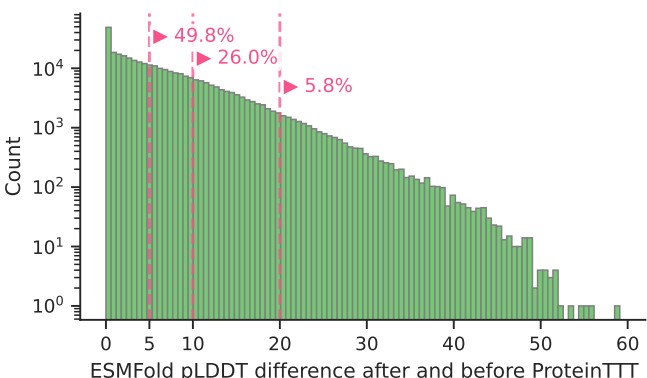

Figure A12: **Magnitude of ESMFold pLDDT improvements after customization with ProteinTTT**. The evaluation is performed on 317,882 proteins from the Big Fantastic Virus Database (BFVD). Percentage annotations indicate the fraction of proteins whose pLDDT increases by at least the corresponding value (e.g., 49.8% of proteins show an improvement of at least 5 pLDDT points).

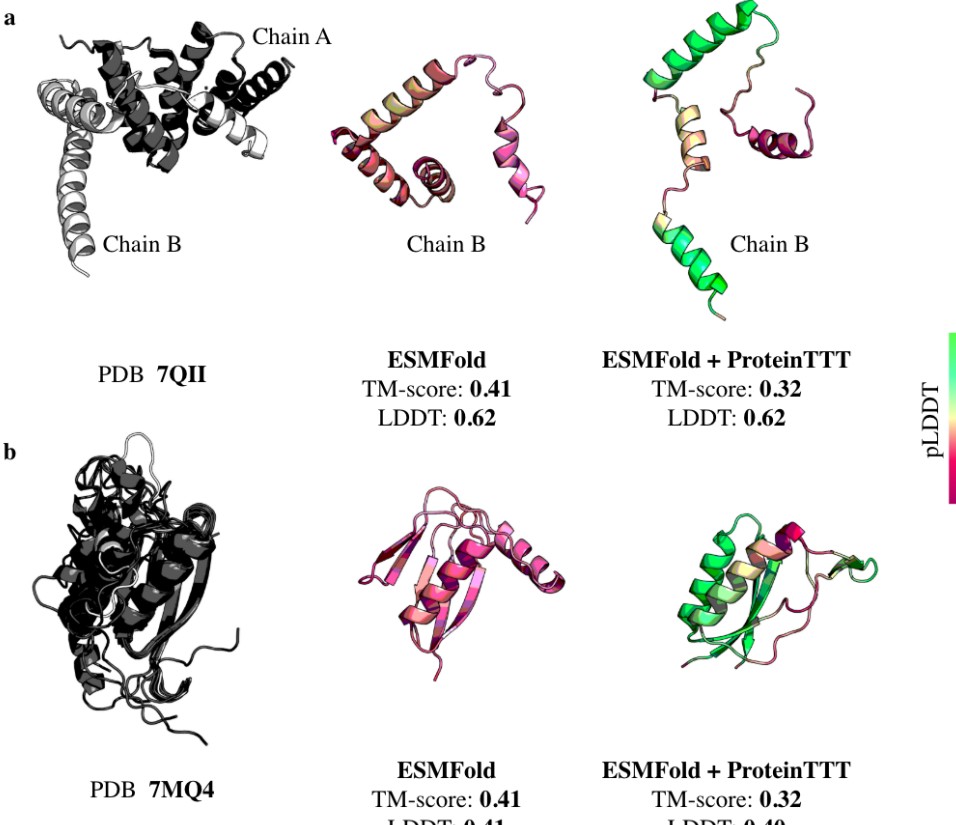

Figure A13: **Detailed analysis of ProteinTTT failure cases on the CAMEO test set**. The figure shows the two entries that consistently exhibit a decrease in TM-score after customization with ProteinTTT across most random seeds (see Figure A6). **(a)** For chain B of PDB entry 7QII (white), the ground-truth structure is part of a dimer in which the conformation of chain B depends on interactions with chain A (black). In the monomeric prediction setting, this context is absent, making the precise helix arrangement inherently ambiguous. Both ESMFold and ESMFold + ProteinTTT correctly capture the helical composition but differ in the global configuration, leading to different TM-scores. **(b)** For chain A of PDB entry 7MQ4 (white), the reference structure is an NMR ensemble with substantial conformational variability (black). Both ESMFold and ESMFold + ProteinTTT recover the stable substructure (right part of the structure in black consisting of a helix surrounded by beta strands), yet produce different conformations in the flexible regions, where multiple arrangements are plausible.

