# OpenReview forum: "One protein is all you need"
_ICLR.cc/2026/Conference — ICLR 2026 Poster_

### Official Review · Reviewer_wNTv · 2025-10-16

**Soundness:** 4
**Presentation:** 3
**Contribution:** 3
**Rating:** 6
**Confidence:** 4

**Summary:**

This paper introduces Protein Test-Time Training (ProteinTTT), a novel method for customizing pre-trained protein language models (PLMs) for individual protein sequences during inference. The method improves performance across protein structure, fitness, and function prediction tasks, especially for proteins with limited homologous sequences. This motivation is clear and well-supported: in most scenarios, researchers are primarily interested in single or a few protein sequences, and ProteinTTT is able to enhance this. Extensive experiments demonstrate that ProteinTTT is transplantable across diverse PLM backbones.

**Strengths:**

* The idea of introducing test time training to PLMs is interesting and convincing. Some case studies also confirm the validity of this method when existing methods are unable to solve specific protein sequences.
* The experiment is rigorously conducted. ProteinTTT is tested across diverse downstream tasks and various backbones.
* By enabling more accurate predictions for individual proteins, ProteinTTT could significantly advance areas such as personalized medicine, antibody design, and protein engineering, where accurate, protein-specific models are essential.

**Weaknesses:**

* As shown in Fig. A4, the time consumption caused by ProteinTTT is non-trivial, especially when pLDDT is calculated. However, the performance improvement seems trivial in Tab. 2, which is my primary concern. Despite some case studies showing that ProteinTTT can improve significantly, the overall performance does not show similar results.

* While the results are strong on average, the paper could benefit from a deeper analysis of the cases where ProteinTTT fails or degrades performance. Figure A6 indicates that for a small fraction of proteins, the prediction quality worsens after customization.

* Simply get better representation from backbones, and freezing the prediction heads is kind of confusing. Modifying the distribution of representation without aligning it to the prediction heads is somehow irrational.

**Questions:**

* Can you provide a percentage of improved samples in Tab. 1 and 2? Ideally, most of the samples will be improved over backbones by test time training.
* If not all samples are improved, can you further analysis the failure cases?
* For fitness and function prediction, how was the number of customization steps (T=30) determined, and how sensitive are the final results to this choice?

---

> ### Author Response · Authors · 2025-11-25
>
> We appreciate the reviewer’s thorough and valuable comments, and we address each of them below.
>
> Weaknesses:
> > As shown in Fig. A4, the time consumption caused by ProteinTTT is non-trivial, especially when pLDDT is calculated. However, the performance improvement seems trivial in Tab. 2, which is my primary concern. Despite some case studies showing that ProteinTTT can improve significantly, the overall performance does not show similar results.
>
> While ProteinTTT yields substantial improvements in protein structure prediction, we acknowledge that its gains in fitness prediction (Table 2\) may appear modest, though still **statistically significant** (for example, the differences between the most performant ProSST and ProSST \+ ProteinTTT have p \< 0.05 according to a paired t-test). We hypothesize that this is due to **saturation of the benchmark itself**, consistent with recent observations ([Notin, 2025](https://pascalnotin.substack.com/p/have-we-hit-the-scaling-wall-for)). Another potential reason for saturated performance is that the proteins in ProteinGym, having rich deep mutational scanning data, may be well represented in the training data of protein language models, whereas **ProteinTTT works best for poorly represented, out-of-distribution sequences** (Figure 5c). This interpretation is also supported by the version of Table 2 split by MSA depth (Table A4), which shows that ProteinTTT tends to improve the fitness prediction performance of base models primarily for proteins with low MSA depth, but offers little improvement or even slightly deteriorates performance when MSA depth is medium or high.
>
> In future work, we plan to apply ProteinTTT to predict the fitness of viral sequences, which is known to be an important ([Hie et al., 2021](https://www.science.org/doi/10.1126/science.abd7331)) yet very hard problem ([Gurev et al., 2025](https://www.biorxiv.org/content/10.1101/2025.08.04.668549v1)). We expect ProteinTTT to deliver more tangible improvements in this setting, similar to the gains observed for viral protein structure prediction, as viral proteins often lack close homologs in sequence databases.
>
> > While the results are strong on average, the paper could benefit from a deeper analysis of the cases where ProteinTTT fails or degrades performance. Figure A6 indicates that for a small fraction of proteins, the prediction quality worsens after customization.
>
> We thank the reviewer for raising this important point. **We analyzed the two proteins from Figure A6 where ProteinTTT worsens prediction quality in the new Figure A13 discussed in the new Appendix H**. We found that the **performance degradation can be explained by the ambiguity of evaluation itself**. Specifically, the figure shows that one of the ground-truth proteins is a dimer, while the prediction is made from a monomer sequence. The second ground truth structure is an NMR ensemble with multiple plausible conformations. Unfortunately, these examples illustrate the challenge of identifying a general reason for the occasional degradation of performance.
>
> For the fitness prediction results, the failure cases are harder to interpret, but we consistently observe that **ProteinTTT tends to increase the performance for sequences poorly represented in the training data and preserve or occasionally decrease performance for sequences that are represented well**, as described in our response to the previous point and shown in Table A4. Interestingly, for protein structure prediction we observe a related trend that depends on the number of similar sequences available in protein databases: the pLDDT improvement tends to decrease as the number of similar sequences increases (Figure 5c).

---

> > ### Author Response · Authors · 2025-11-25
> >
> > Questions:
> > > Can you provide a percentage of improved samples in Tab. 1 and 2? Ideally, most of the samples will be improved over backbones by test time training.
> >
> > **Indeed, most of the samples are improved upon using ProteinTTT for all methods evaluated in Tab. 1 and 2\.** We provide the percentages in the corresponding Tables R4 and R5 below.
> >
> > Table R4. Percentage of improved samples with ProteinTTT for structure prediction (extension of Table 1).
> >
> > | Method | Structures with improved LDDT upon ProteinTTT |
> > | :---- | :---- |
> > | ESM3 | 61% |
> > | HelixFold-Single | 50% |
> > | ESMFold | 67% |
> >
> > Table R5. Percentage of improved samples upon ProteinTTT for fitness prediction (extension of Table 2, including newly added ProGen2 discussed in detail in response to the reviewer DpSs).
> >
> > | Method | DMS experiments with improved Spearman correlation upon ProteinTTT |
> > | :---- | :---- |
> > | ESM2 (35M)  | 77% |
> > | ProGen2-small (151M) | 58% |
> > | SaProt (35M)  | 62% |
> > | ESM2 (650M)  | 67% |
> > | SaProt (650M)  | 53% |
> > | ProSST (K=2048)  | 56% |
> >
> > > If not all samples are improved, can you further analysis the failure cases?
> >
> > Please see our response to the second Weakness point above for the analysis of the failure cases.
> >
> > > For fitness and function prediction, how was the number of customization steps (T=30) determined, and how sensitive are the final results to this choice?
> >
> > We thank the reviewer for raising this important point. Indeed, as shown in the hyperparameters Table A3, we did not explore different numbers of steps. This is because the **ProteinTTT configuration effectively has two degrees of freedom despite having three hyperparameters**: the number of steps, the learning rate, and the batch size (or the number of gradient-accumulation steps). In other words, with more aggressive learning-rate or batch-size configurations, ProteinTTT can reach optimal performance in fewer steps. Figure A10 shows the sensitivity of ProteinTTT to the learning rate and batch size across different step counts on fitness prediction. We began our experiments with 30 steps and kept this choice throughout to maintain consistency.
> >
> > Figure A7 additionally demonstrates the sensitivity of the final results on the CAMEO structure prediction test set with respect to the number of customization steps. The figure shows that **the highest improvement is achieved during the first customization steps** and plateaus later. This observation also suggests that the computational cost of ProteinTTT can be reduced by lowering the number of steps for the price of slightly lower performance.
> >
> > We are happy to answer any additional or unclear points.

---

> > > ### Comment · Reviewer_wNTv · 2025-11-28
> > > **Response**
> > >
> > > Seems fine to me. Most proteins can be improved by ProteinTTT. I decide to keep the score.

---

### Official Review · Reviewer_MtJ2 · 2025-10-20

**Soundness:** 3
**Presentation:** 4
**Contribution:** 2
**Rating:** 4
**Confidence:** 3

**Summary:**

The authors propose ProteinTTT, a method to adapt protein foundation models like AlphaFold2 and ESMFold to specific new proteins at test time. In essence, it involves running a few steps of self-supervised fine-tuning of the main trunks of the model in question independently of any task-specific heads trained on top of them. The authors show that this procedure improves performance on a variety of downstream tasks like folding and fitness prediction, especially for proteins without known homologues.

**Strengths:**

The manuscript is exceptionally clearly written, and the method is intuitive and easy to follow. The authors include a diverse set of experiments run on several different models, and I appreciate the inclusion of confidence intervals throughout.

Methods like ProteinTTT obviously have long pedigrees in the language modeling and computer vision literature, but this is the first paper I'm aware of that applies this sort of idea to protein modeling.

**Weaknesses:**

The most glaring weakness in my eyes is that the effect of adaptation seems to be relatively limited. On several benchmarks, improvements are measured as fractions of percentage points. On several others (especially in section 5), the improvement is more difficult to gauge; at various points, the authors make claims about the fractions of proteins that are improved by the method, but details about the magnitude of the improvement in these cases seem to be pretty sparse. Figure 5b hints that some proteins do indeed see large improvements, but it's also difficult to read. Some more clarity here would be appreciated.

Some other assorted issues/potential areas of improvement:

1. The authors occasionally use pLDDT of the tuned model (e.g. to choose the number of optimization steps), which seems a little problematic to me; since you're training on one protein, you're probably also inflating confidence on that protein. Could the authors validate that this isn't happening, perhaps by running early stopping experiments that "cheat" by observing downstream metrics? Another thing that would be good to see would be something like Figure 5b, but pitting ESMFold + ProteinTTT LDDT against ESMFold + ProteinTTT pLDDT
2. Is there a reason the authors only test 18 CAMEO proteins? How does the method perform on proteins that aren't extremely low-confidence?
3. Sometimes, the text overstates the results. For example: "As shown in Figure 4a, ESMFold + ProteinTTT achieves significantly
higher average LDDT scores compared to general-purpose ESMFold." The improvement in this case is not actually "significantly higher" in the statistical sense, since the error bars overlap.

**Questions:**

See above.

---

> ### Author Response · Authors · 2025-11-20
>
> We appreciate the important comments of the reviewer and provide clarifications for each of them below, supported by new experiments.
>
> > The most glaring weakness in my eyes is that the effect of adaptation seems to be relatively limited. On several benchmarks, improvements are measured as fractions of percentage points. On several others (especially in section 5), the improvement is more difficult to gauge; at various points, the authors make claims about the fractions of proteins that are improved by the method, but details about the magnitude of the improvement in these cases seem to be pretty sparse. Figure 5b hints that some proteins do indeed see large improvements, but it's also difficult to read. Some more clarity here would be appreciated.
>
> Thank you for raising this issue. This is an important point, and we are glad to provide additional clarifications, which we will also incorporate into the manuscript. Across all benchmarks, we consistently report the average performance before and after ProteinTTT to illustrate the effect of adaptation. This includes Table 1 for structure prediction on CAMEO, Table 2 for fitness prediction on ProteinGym, and Figure 4 for antibody/antigen structure prediction on SAbDab. In all these cases, **the magnitude of improvement is statistically significant** for the best-performing models.
>
> In the BFVD case study in Section 5, where ground-truth structures are not available, we measure improvement before and after customization relative to AlphaFold2. **To further clarify the magnitude of improvements shown in Figure 5 in Section 5, we now include a new Figure A12**, which shows the distribution of pLDDT improvements from ProteinTTT adaptation. This figure shows that 50% of proteins (158K) have an improvement of at least 5 pLDDT points, and, for example, 26% of proteins (83K) show an improvement of at least 10 pLDDT points.
>
> Please let us know if any specific points remain unclear or if further explanation would be helpful.
>
> > 1\. The authors occasionally use pLDDT of the tuned model (e.g. to choose the number of optimization steps), which seems a little problematic to me; since you're training on one protein, you're probably also inflating confidence on that protein. Could the authors validate that this isn't happening, perhaps by running early stopping experiments that "cheat" by observing downstream metrics? Another thing that would be good to see would be something like Figure 5b, but pitting ESMFold \+ ProteinTTT LDDT against ESMFold \+ ProteinTTT pLDDT
>
> We fully understand the reviewer’s concern. For this reason, **we validated that ProteinTTT does not simply inflate confidence scores but also improves actual downstream performance, with gains that correspond to the increases in confidence**. Figure A9 presents a plot addressing the suggestion of the reviewer. It shows that the Pearson correlation between pLDDT and LDDT is 0.875 across 18K structures from BFVD (all structures with AlphaFold2 pLDDT \> 90, where the computation of LDDT is reasonable).
>
> **We also implemented the “cheating” (oracle) experiment suggested by the reviewer** to illustrate that selecting the optimal step for each protein based on confidence (pLDDT) approaches the performance achieved when selecting the optimal step using the true LDDT (“cheating”/oracle), and is clearly better than selecting steps at random. Specifically, Table R3 below shows that:
>
> 1. Even selecting a step at random (“ESMFold \+ ProteinTTT (random step)”) improves over vanilla ESMFold (without ProteinTTT), suggesting that customization via minimizing perplexity is a right direction for enhancing downstream structure prediction.
> 2. Our approach (“ESMFold \+ ProteinTTT (max pLDDT step)”) performs even better, approaching the oracle performance obtained when selecting the step using the true LDDT (“ESMFold \+ ProteinTTT (max LDDT step)”).
> 3. Interestingly, our approach also outperforms selecting the single globally optimal step across all proteins via oracle and using it for early stopping (“ESMFold \+ ProteinTTT (global max LDDT step)”), suggesting that per-protein pLDDT–based early stopping is more effective than using a single best step shared across proteins.
>
> Table R3. Structure prediction performance on the CAMEO test set. The table reports the average LDDT. For efficiency during the rebuttal, we use only a single random seed for ProteinTTT.
>
> | Method | LDDT ↑ |
> | ----: | ----: |
> | ESMFold | 52.27 |
> | ESMFold \+ ProteinTTT (random step, seed=1) | 52.66 |
> | ESMFold \+ ProteinTTT (random step, seed=4) | 52.80 |
> | ESMFold \+ ProteinTTT (random step, seed=2) | 53.06 |
> | ESMFold \+ ProteinTTT (random step, seed=3) | 53.06 |
> | ESMFold \+ ProteinTTT (random step, seed=0) | 53.26 |
> | ESMFold \+ ProteinTTT (global max LDDT step; global Oracle) | 53.93 |
> | ESMFold \+ ProteinTTT (max pLDDT step; Ours) | 55.37 |
> | ESMFold \+ ProteinTTT (max LDDT step; pre-protein Oracle) | 57.81 |

---

> > ### Author Response · Authors · 2025-11-20
> >
> > > 2\. Is there a reason the authors only test 18 CAMEO proteins? How does the method perform on proteins that aren't extremely low-confidence?
> >
> > The 18 CAMEO proteins were selected as the proteins that are already not predicted well by ESMFold (pLDDT \< 70 and perplexity \> 6; standard criteria for filtering out high-confidence predictions ([Lin et al., 2023](https://www.science.org/doi/10.1126/science.ade2574))). **The rest of the CAMEO set is less relevant for benchmarking test-time customization, as the predictions are already of high quality**, and customization at best either preserves performance or yields only slight improvements. Indeed, only 2 out of the remaining 144 proteins show a substantial increase in LDDT and TM-score (\> 0.05) after applying ProteinTTT, illustrating that customization may be redundant for already well-predicted structures. This approach of applying test-time customization only to non-high-confidence predictions can be viewed as “compute-optimal test-time scaling” ([Snell et al., 2024](https://arxiv.org/abs/2408.03314)).
> >
> > Nevertheless, to achieve the highest possible performance on predicting viral protein structures in our case study, **we have now applied ProteinTTT to the entire BFVD database, including proteins with ESMFold pLDDT \> 70** (please see the updated text in Section 5 in blue). As expected, the number of AlphaFold2 predictions improved by ESMFold \+ ProteinTTT increased, but more moderately than for the non-high-confidence half of the database. The proportion of improved structures rose from 17% to 19% (an increase of additional 2%), corresponding to 61,308 improved structures in total.
> >
> > > 3\. Sometimes, the text overstates the results. For example: "As shown in Figure 4a, ESMFold \+ ProteinTTT achieves significantly higher average LDDT scores compared to general-purpose ESMFold." The improvement in this case is not actually "significantly higher" in the statistical sense, since the error bars overlap.
> >
> > We apologize for the confusion. In fact, these results **are "significantly higher" in the statistical sense** (p-value of a paired t-test \< 0.05 in both cases). By mistake, we did not include the information about statistical tests in the text, and the conclusion indeed looked overstated. We have updated the mentioned sentence accordingly: *“... ESMFold \+ ProteinTTT achieves significantly higher average LDDT scores compared to general-purpose ESMFold \\updated{(paired t-test p-value \< 0.05)}”*.
> >
> > Please let us know if there are other parts of the text where results seem to be overstated.

---

> > > ### Comment · Reviewer_MtJ2 · 2025-11-25
> > >
> > > This all looks good to me. Raising from (4 -> 6).

---

### Official Review · Reviewer_9CBZ · 2025-10-31

**Soundness:** 4
**Presentation:** 4
**Contribution:** 3
**Rating:** 8
**Confidence:** 3

**Summary:**

This paper proposes ProteinTTT, a test-time training framework aimed at improving the downstream performance of protein language models. Motivated by the observation that sequence-level perplexity correlates with task accuracy and that low performing cases often exhibit high perplexity, the authors finetune the pretrained model at test time using the perplexity of each target sequence. The method is evaluated across three downstream tasks and several case studies, demonstrating consistent improvements across both tasks and backbone protein language models. The results suggest that ProteinTTT can effectively enhance existing protein language models without requiring large-scale retraining or additional labeled data.

**Strengths:**

1. The manuscript is well written and polished; clear, concise, and detailed enough in both method and evaluation to be easy to follow.
2. The experimental design is comprehensive, covering three relevant downstream tasks with appropriate baselines. The inclusion of case studies and in-depth analyses further strengthens the empirical validation.
3. The results are consistent and convincing, showing that ProteinTTT can effectively enhance the performance of existing protein language models.
4. Introducing test-time training to protein language models, and framing perplexity minimization as a way to adapt to distribution shifts in downstream tasks, offers a novel and insightful perspective.

**Weaknesses:**

1. The proposed method seems inherently limited in scope, it applies only to protein language models and only to sequence-based tasks. This restricts its applicability to broader classes of protein modeling problems.
2. While the reported improvements are consistent across benchmarks, many gains are modest and appear bounded by the baseline model’s performance. The authors briefly acknowledge this (e.g., in G1), but a clearer discussion in the main text along with analysis of failure cases would help readers better understand when and why ProteinTTT is most effective.
3. Not necessarily a weakness, but the title "One protein is all you need" feels somewhat overstated. ProteinTTT indeed brings measurable benefits, yet its performance remains constrained by the underlying model’s capacity and generalizability. That said, the title is understandable as a stylistic choice.

**Questions:**

**(A) Scope and Applicability.**

 1. Since ProteinTTT relies on self-supervised fine-tuning over sequence perplexity, it appears primarily suited for tasks that take protein sequences as input and for models trained with language modeling objectives. This seems limit its direct applicability to sequence-based models rather than structure-based ones (e.g., AlphaFold3) which usually have stronger performance.
 2. Could the authors discuss whether ProteinTTT can be extended to other task types beyond the three evaluated in this work, such as sequence design?

**(B) Performance on Structure Prediction.**
1. Can authors add the performance of the current any state-of-the-art results on the CAMEO benchmark (e.g., AlphaFold2, AlphaFold3, Boltz) as a reference?
2. The recently proposed DPLM-BiT model (https://arxiv.org/abs/2504.11454) demonstrates strong folding performance which also uses a masked protein language modeling approach. Including it as a baseline could provide a more rigorous comparison and help assess whether ProteinTTT maintains its advantage on this class of models.
3. Has the method been explored for more challenging structure prediction tasks, such as protein–protein complex prediction?

**(C) Experimental Details and Behavior.**
1. In the customization steps (e.g., Fig. 3), would continued test-time fine-tuning eventually lead to overfitting or catastrophic forgetting?
2. How many fine-tuning steps are typically recommended for each task, and are these consistent across tasks?
3. Can the authors discuss more on the conditions where ProteinTTT might fail? For test time training, an extended discussion on failure cases or scenarios would provide valuable practical guidance.

---

> ### Author Response · Authors · 2025-11-28
>
> We appreciate the reviewer’s careful and valuable feedback, and we address each point below. We apologize for the delayed response.
>
> Weaknesses
>
> > 1\. The proposed method seems inherently limited in scope, it applies only to protein language models and only to sequence-based tasks. This restricts its applicability to broader classes of protein modeling problems.
>
> We thank the reviewer for raising this point. Part of our initial motivation for developing self-supervised customization for proteins was the widespread utility of foundation protein language models, which has enabled a paradigm shift in computational biology. Particularly illustrative examples are the ESM3 model as well as DPLM-BiT, newly added to our experiments, which can be applied effectively across nearly all major protein machine learning tasks, including but not limited to protein structure and sequence design, inverse folding, or protein function prediction. While we currently demonstrate the applicability of ProteinTTT to ESM3 and DPLM-BiT only in the context of structure prediction, the same approach can be straightforwardly extended to other tasks by performing self-supervised customization on the corresponding input tracks of these multimodal models (sequence, structure, or function tokens).
>
> Ultimately, our method is conceptually not limited to protein language models and can, in principle, be applied to other architectures such as AlphaFold2 or BoltzGen. **We have now added a new section, “Extension to other model types and tasks,” in Appendix H (“Limitations and future work”) to elaborate on these points**.
>
> > 2\. While the reported improvements are consistent across benchmarks, many gains are modest and appear bounded by the baseline model’s performance. The authors briefly acknowledge this (e.g., in G1), but a clearer discussion in the main text along with analysis of failure cases would help readers better understand when and why ProteinTTT is most effective.
>
> We thank the reviewer for raising this important point. **We analyzed the structure prediction failure cases in detail in the new Figure A13 discussed in the new paragraph “Better control over failure cases” of Appendix H**. We found that in these cases, the ground-truth structures are ambiguous given the input chain sequences. Specifically, the figure shows that one of the ground-truth proteins is a dimer, while the prediction is made from a monomer sequence. The second ground truth structure is an NMR ensemble with multiple plausible conformations. Unfortunately, these examples illustrate the challenge of identifying a general reason for the occasional degradation of performance.
>
> More generally, while it is difficult to identify a general trend for success and failure modes, we consistently observe that **ProteinTTT tends to increase performance for sequences that are poorly represented in the training data, and to preserve or occasionally slightly decrease the performance for sequences that are well represented**. For fitness prediction, this is shown in Table A4, which demonstrates that ProteinTTT tends to improve the performance of the base models primarily for proteins with low MSA depth (approximating the number of similar examples in the training data), but offers little improvement or can even deteriorate performance when MSA depth is medium or high. For protein structure prediction, we observe a related trend that depends on the number of similar sequences available in protein databases: the pLDDT improvement tends to decrease as MSA depth increases (Figure 5c).
>
> This observation may also partially explain why the improvements with ProteinTTT are more pronounced for structure prediction of viral proteins compared to, for example, fitness prediction on ProteinGym (**where improvements are indeed modest though statistically significant**; for example, the differences between the most performant ProSST and ProSST \+ ProteinTTT have p \< 0.05 according to a paired t-test). ProteinGym, with its rich deep mutational scanning data, is likely well represented in the training data of protein language models, whereas viral proteins often lack homologs in UniProt. As a result, ProteinTTT can yield substantial improvement on viral proteins that is not bounded by the baseline model’s performance (i.e., **ESMFold+ProteinTTT outperforms AlphaFold2 on 19% of viral proteins, even though ESMFold alone outperforms AlphaFold2 on only 10% of cases, as shown in Figure 5a**).

---

> > ### Author Response · Authors · 2025-11-28
> >
> > Questions
> >
> > > (A) Scope and Applicability.
> >
> > > 1. Since ProteinTTT relies on self-supervised fine-tuning over sequence perplexity, it appears primarily suited for tasks that take protein sequences as input and for models trained with language modeling objectives. This seems limit its direct applicability to sequence-based models rather than structure-based ones (e.g., AlphaFold3) which usually have stronger performance.
> >
> > Please see our response to Weakness 1 and the newly added section “Extension to other model types and tasks” in the new Appendix H “Limitations and future work”, where we discuss how **ProteinTTT could be extended beyond sequence-based models, including application to AlphaFold3**.
> >
> > > 2. Could the authors discuss whether ProteinTTT can be extended to other task types beyond the three evaluated in this work, such as sequence design?
> >
> > Indeed, extending ProteinTTT to protein design applications is an exciting direction for future work. We **have now implemented ProteinTTT for the generative/autoregressive ProGen2 model for sequence design** and demonstrated its effectiveness on fitness prediction ([our first response to reviewer DpSs](https://openreview.net/forum?id=5zAde2jch7&noteId=Fmm9WqlSHz) for details). These **results suggest that ProteinTTT could also improve sequence design**. In addition, section “Extension to other model types and tasks” in Appendix H “Limitations and future work”, now also discusses the application of ProteinTTT to other tasks, including sequence design within *de novo* protein design.
> >
> > > (B) Performance on Structure Prediction.
> >
> > > 1. Can authors add the performance of the current any state-of-the-art results on the CAMEO benchmark (e.g., AlphaFold2, AlphaFold3, Boltz) as a reference?
> >
> > We have now added AlphaFold2 results and report them in Table R6 below (we have chosen AlphaFold2 for consistency with the viral protein structure prediction experiments). As expected, AlphaFold2 on average outperforms both ESMFold and ESMFold \+ ProteinTTT. Please note, however, that **this comparison is not entirely fair, as our CAMEO test set was constructed to specifically include the most challenging examples for ESMFold** (pLDDT \< 70 and perplexity \> 6\) to benchmark customization, whereas some of these entries can be solved well using AlphaFold2 (7 out of 18 examples have AlphaFold2 pLDDT \> 90). Nevertheless, even in this setup ESMFold outperforms AlphaFold2 in 2 out of 18 cases, and ESMFold \+ ProteinTTT outperforms AlphaFold2 in 4 out of 18 cases (Table R7). In other words, **ProteinTTT enables ESMFold to additionally surpass AlphaFold2 in 2 of these 18 examples** despite not using MSA information. This is further expanded in our experiments on **viral protein structure prediction (Section 5.2), where ESMFold+ProteinTTT improves 19% (61,308) structures predicted with AlphaFold2**.
> >
> > Table R6. Structure prediction results including newly added AlphaFold2 ([Jumper et al., 2021](https://www.nature.com/articles/s41586-021-03819-2)) and DPLM2 Bit-based ([Hsieh et al., 2025](https://arxiv.org/abs/2504.11454v3)) models. The metrics show TM-score and LDDT metrics averaged across 18 ESMFold low-confidence targets in the CAMEO test set (extension of Table 1).
> >
> > | Method                                   | TM-score ↑ | LDDT ↑  |
> > |------------------------------------------|------------|---------|
> > | ESM3 (Hayes et al., 2024)                | 0.3480     | 0.3723  |
> > | ESM3 + CoT (Hayes et al., 2024)          | 0.3677     | 0.3835  |
> > | ESM3 + ProteinTTT (Ours)                 | **0.3954** | **0.4214** |
> > | | | |
> > | DPLM-BiT (Hsieh et al., 2025)            | 0.3592     | 0.4598  |
> > | DPLM-BiT + ProteinTTT (Ours)             | **0.3813** | **0.4799** |
> > | | | |
> > | HelixFold-Single (Fang et al., 2023)     | 0.4709     | 0.4758  |
> > | HelixFold-Single + ProteinTTT (Ours)     | **0.4839** | **0.4840** |
> > | | | |
> > | ESMFold (Lin et al., 2023)               | 0.4649     | 0.5194  |
> > | ESMFold + MP (Lin et al., 2023)          | 0.4862     | 0.5375  |
> > | ESMFold + ProteinTTT (Ours)              | **0.5047** | **0.5478** |
> > | | | |
> > | AlphaFold2 w/o MSA (Jumper et al., 2021) | 0.3159     | 0.4235  |
> > | AlphaFold2 (Jumper et al., 2021)         | 0.7305     | 0.7637  |
> >
> > Table R7. Detailed results based on Table R6, highlighting the performance of ESMFold \+ ProteinTTT on 4 out of 18 CAMEO test set entries where it outperforms AlphaFold2.
> >
> > | Model                               | TM-score ↑ |        |        |        |
> > |-------------------------------------|------------|--------|--------|--------|
> > |                                     | 7zro\_A     | 7oa7\_A | 7qao\_A | 7ri3\_C |
> > | AlphaFold2 (Jumper et al., 2021\)    | 0.21145    | 0.71966 | 0.36698 | 0.30149 |
> > | ESMFold (Lin et al., 2023\)          | **0.24755**    | 0.67148 | 0.31395 | 0.34854 |
> > | ESMFold \+ ProteinTTT (Ours)         | **0.24755**    | **0.74021** | **0.40052** | **0.40771** |

---

> > > ### Author Response · Authors · 2025-11-28
> > >
> > > > 2. The recently proposed DPLM-BiT model ([https://arxiv.org/abs/2504.11454](https://arxiv.org/abs/2504.11454)) demonstrates strong folding performance which also uses a masked protein language modeling approach. Including it as a baseline could provide a more rigorous comparison and help assess whether ProteinTTT maintains its advantage on this class of models.
> > >
> > >    We appreciate the reviewer’s suggestion, and we have implemented DPLM-BiT \+ ProteinTTT based on DPLM-BiT ([Hsieh et al., 2025](https://arxiv.org/abs/2504.11454v3); also referred to as DPLM2 Bit-based in the original publication). **Table R6 above demonstrates that ProteinTTT improves the performance of DPLM-BiT** (statistically significant with a paired t-test p-value <0.05).
> > >
> > >    Please note that the current version of DPLM-BiT cannot fully leverage ProteinTTT, as it does not implement pLDDT prediction or any other confidence function (we observe that pLDDT predictions for all residues are roughly 50±2 in both DPLM2 Bit-based and DPLM2, so we assume the pLDDT head was not trained in the currently available models). Nevertheless, **despite the absence of a confidence function in DPLM-BiT, DPLM-BiT \+ ProteinTTT still shows higher performance than the uncustomized version.** We tune the ProteinTTT hyperparameters in the same way as for other models but use 10 steps (instead of the usual 30\) for efficiency during the rebuttal, and based on our new insights suggesting that a lower number of steps may be sufficient (e.g., Figure A7, Figure A10). The final hyperparameters are: 10 steps, batch size 2, gradient-accumulation steps 4, learning rate 4e-6.
> > >
> > >    While DPLM-BiT is a discrete diffusion model, we performed ProteinTTT customization using the standard BERT training objective (i.e., a fixed 15% masking ratio instead of sampling different ratios according to a diffusion schedule). Nevertheless, **the observed improvements demonstrate that even a simple fixed masking ratio may be sufficient for customizing discrete diffusion protein language models.** Experimenting with training objectives tailored for customizing DPLM-BiT (and other discrete diffusion models) presents an interesting future work direction.
> > >
> > >    The application of ProteinTTT to DPLM-BiT is particularly exciting because it makes our customization approach directly applicable to multiple new tasks, such as inverse folding (via customization to structural tokens) or multimer structure prediction (via customization to several sequences at once). This opens promising avenues for future research.
> > >
> > > > 3. Has the method been explored for more challenging structure prediction tasks, such as protein–protein complex prediction?
> > >
> > > While we did not experiment with protein–protein complex prediction in the current work, ProteinTTT shows promise for improving performance on this task as well. As discussed in the previous point, for example, ProteinTTT could be directly applied to protein–protein complex prediction through the customization of DPLM-BiT. The consistent improvements with ProteinTTT across different models and tasks suggest that DPLM-BiT + ProteinTTT could similarly enhance the performance of DPLM-BiT on this task.
> > >
> > >    Additionally, in the new paragraph “Extension to other model types and tasks” in Appendix H, we describe how ProteinTTT could be extended to models such as AlphaFold-Multimer or AlphaFold3/Boltz for protein–protein complex prediction and beyond.
> > >
> > > > (C) Experimental Details and Behavior.
> > >
> > > > 1. In the customization steps (e.g., Fig. 3), would continued test-time fine-tuning eventually lead to overfitting or catastrophic forgetting?
> > >
> > >    **Indeed, customization eventually leads to overfitting, but this can be detected using the pLDDT confidence function.** To illustrate this, we repeat the experiment from Fig. 3 for 100 steps (instead of the usual 30\) using five different random seeds. Across all five seeds, ProteinTTT consistently finds the best-pLDDT solution within the first 20 steps (pLDDT 39-\>80 at step 4, 39-\>50 at step 18, 39-\>77 at step 4, 39-\>78 at step 7, 39-\>80 at step 5\) and, after roughly step 30, plateaus at a decreased pLDDT (final-step pLDDT values: 48, 42, 38, 41, 43). Selecting the step with the highest pLDDT leads to improved performance as measured by LDDT or TM-score, as shown in Fig. 3, effectively enabling overcoming overfitting. Interestingly, even after 100 steps we do not observe catastrophic forgetting, as the final-step pLDDT values remain reasonably high and comparable to those obtained before customization (48, 42, 38, 41, 43 versus 39 in this case).

---

> > > > ### Author Response · Authors · 2025-11-28
> > > >
> > > > > 2. How many fine-tuning steps are typically recommended for each task, and are these consistent across tasks?
> > > >
> > > >    This is an important point which needs further clarification. Even though ProteinTTT has three hyperparameters (the number of steps, learning rate, and batch size / the number of gradient-accumulation steps), they effectively have only two degrees of freedom. In other words, with more aggressive learning-rate or batch-size configurations, ProteinTTT can reach optimal performance in fewer steps. Figure A10 shows the sensitivity of ProteinTTT to the learning rate and batch size across different step counts on fitness prediction. Figure A7 additionally demonstrates the sensitivity of the final results on the CAMEO test set with respect to the number of customization steps. The figure shows that **the highest improvement is achieved during the first customization steps** and plateaus later. This observation also suggests that the computational cost of ProteinTTT can be reduced by lowering the number of steps for the price of slightly lower performance.
> > > >
> > > >    In nearly all our experiments, we **fixed the number of steps to 30 for consistency** and tuned the learning rate and batch size. In conclusion, **we recommend setting the number of steps to 30, following our experiments, or choosing a smaller number of steps if improved computational efficiency is needed.** Choosing a smaller number of steps may also require additional ProteinTTT hyperparameter tuning to achieve the best possible performance.
> > > >
> > > > > 3. Can the authors discuss more on the conditions where ProteinTTT might fail? For test time training, an extended discussion on failure cases or scenarios would provide valuable practical guidance.
> > > >
> > > >    **We have now added a new section, “Better control over failure cases”, in Appendix H (“Limitations and future work”) to provide better practical guidance**. Please also see our response to Weakness 2 above for details.

---

### Official Review · Reviewer_DpSs · 2025-10-31

**Soundness:** 3
**Presentation:** 3
**Contribution:** 2
**Rating:** 4
**Confidence:** 4

**Summary:**

This paper introduces Protein Test-Time Training (ProteinTTT), a self-supervised framework that customized pretrained protein language model to individual target proteins during interence. ProteinTTT fixed the downstream task head and finetuned the model backbone. Experiments show that this consistently improves model performance in different tasks with different kinds of models.

**Strengths:**

1. The method is simple and general. It does not require additional data and can be easily applied to different pretrained PLMs. And the appraoch leverages lightweight finetuning (LoRA and limited steps), making it resource-efficient.
2. The method consistently improve the performance on different tasks and different models.
3. The authors use case studies to demonstrate real-world applications. For biologists and chemists who are interested in specific proteins, this could be useful.

**Weaknesses:**

1. The authors claimed that the method can be easily extended to models with autoregressive masking, while they did not do that. I suggest the authors might try to show the results on ProGen.
2. The improvement on larger models are marginal. While ProteinTTT consistently improve results, the improvement on larger model is marginal and even negligible. For example, in table 2 when it comes to 650M model. The improvement is nearly negligible. I wonder what will happen in larger model like 1B or 3B.
3. There are already a lot of inference scaling ideas rather than test time training in current trends in LLM. It would be valuable for the authors to clarify whether proteinTTT could similarly benefit from such inference-time scaling strategies (e.g. Use MSAs or other context) instead of perform test-time training.

Overall, While the work is technically sound and relevant, its contributions appear more empirical and domain-specific than methodological. The paper might be more impactful if expanded with additional biological validations. In its current form, the scope and framing could align more naturally with a computational biology or bioinformatics venue rather than a general ML conference.

**Questions:**

See weakness

---

> ### Author Response · Authors · 2025-11-20
>
> We appreciate the reviewer’s thorough and valuable comments, and we address each of them below.
>
> > The authors claimed that the method can be easily extended to models with autoregressive masking, while they did not do that. I suggest the authors might try to show the results on ProGen.
>
> This is a valuable suggestion. We implement an autoregressive version of ProteinTTT as suggested by the reviewer and show that **ProteinTTT improves the performance of ProGen2** ([Nijkamp et al., 2023](https://www.cell.com/cell-systems/fulltext/S2405-4712\(23\)00272-7)) on ProteinGym (Table R1). The improvements are statistically significant, with a paired t-test p \< 0.05.
>
> Table R1. Performance of ProteinTTT customization applied to ProGen2-small and ProGen2-large on the ProteinGym benchmark. See Table 2 in the paper for further details of the benchmark. Rows with ProGen2-large are marked with an asterisk (“\*”) because they show results on 191 out of 217 ProteinGym targets. While ProteinTTT itself is computationally cheap (seconds per protein), inference with ProGen2-large on the remaining proteins is computationally expensive (up to 60 hours of GPU compute per protein with hundreds of thousands of mutations) and is currently being computed. We will update the table in the next revision.
>
> | Model | Avg. Spearman ↑ | Spearman by phenotype ↑ |  |  |  |  |
> | ----- | ----- | ----- | ----- | ----- | ----- | ----- |
> | |  | Activity | Binding | Expression | Organismal Fitness | Stability |
> | ProGen2-small (151M) ([Nijkamp et al., 2023](https://www.cell.com/cell-systems/fulltext/S2405-4712\(23\)00272-7)) | 0.3255 | 0.3316 | 0.2681 | 0.3730 | 0.3283 | 0.3264 |
> | ProGen2-small (151M) \+ ProteinTTT (Ours) | **0.3590** | **0.3827** | **0.2959** | **0.3868** | **0.3304** | **0.3992** |
> | ProGen2-large (2.7B) ([Nijkamp et al., 2023](https://www.cell.com/cell-systems/fulltext/S2405-4712\(23\)00272-7))\* | 0.3705 | 0.3948 | 0.2918 | 0.4273 | 0.3653 | 0.3736 |
> | ProGen2-large (2.7B) \+ ProteinTTT (Ours)\* | **0.3822** | **0.4079** | **0.3074** | **0.4326** | **0.3669** | **0.3963** |
>
> To perform single-sequence customization in an autoregressive setting, we apply a standard teacher forcing procedure with a batch size of one. Specifically, each ProteinTTT step optimizes next token prediction across the whole sequence via the following loss function.
>
> \\\[
> \\mathcal{L}\_{\\text{AR}}(x;\\theta)
> \= \\frac{1}{|x|} \\sum\_{i=1}^{|x|}
>     \- \\log p(x\_i \\mid x\_{\< i}; \\theta),
> \\\]
> where \\(x\\) denotes a sequence of protein tokens, and \\(p(x\_i \\mid x\_{\< i}; \\theta) \\doteq g(f(x\_{\< i}; \\theta))\_{x\_i}\\) is the probability assigned by the model to the true token \\(x\_i\\) given all preceding tokens \\(x\_{\< i}\\). We tune the hyperparameters (batch size and learning rate) similarly to the other models in the paper and apply ProteinTTT to ProGen2-large (2.7B parameters) using LoRA, similarly to ESMFold backbone (3B parameters).
>
> > The improvement on larger models are marginal. While ProteinTTT consistently improve results, the improvement on larger model is marginal and even negligible. For example, in table 2 when it comes to 650M model. The improvement is nearly negligible. I wonder what will happen in larger model like 1B or 3B.
>
> This is an important point that requires clarification. We appreciate the reviewer for raising it, and we will revise the text accordingly. We summarize this below.
>
> While ProteinTTT yields substantial improvements in protein structure prediction, we acknowledge that its gains in fitness prediction may appear modest for larger models (though still **statistically significant**; for example, the differences between the most performant ProSST and ProSST \+ ProteinTTT have *p* \< 0.05 according to a paired t-test). We hypothesize that this is due to **saturation of the benchmark itself**, consistent with recent observations [(Notin, 2025\)](https://pascalnotin.substack.com/p/have-we-hit-the-scaling-wall-for). Another potential reason for saturated performance is that the proteins in ProteinGym, having rich deep mutational scanning data, may be well represented in protein language model training data, while ProteinTTT works the best for poorly represented, out-of-distribution sequences (Figure 5c).
>
> Please note that we did not include ESM2 (1B) and ESM2 (3B) in Table 2, as they perform worse on fitness prediction than ESM2 (650M) and are therefore not practically relevant for this task. Nevertheless, **our central results rely on ESMFold, which uses ESM2 with 3B parameters as its backbone** (and 1.4B parameters in the structure prediction head). This demonstrates that ProteinTTT works effectively with larger backbones when downstream performance is not saturated, particularly in challenging scenarios such as predicting antibody CDRs or viral proteins (Section 5).

---

> > ### Author Response · Authors · 2025-11-20
> >
> > > There are already a lot of inference scaling ideas rather than test time training in current trends in LLM. It would be valuable for the authors to clarify whether proteinTTT could similarly benefit from such inference-time scaling strategies (e.g. Use MSAs or other context) instead of perform test-time training.
> >
> > Indeed, recent research on LLMs offers many ideas that can further extend ProteinTTT. Leveraging MSAs is a natural and appealing extension, analogous to using similar sentences at inference time in LLMs ([Hübotter et al., 2025](https://arxiv.org/abs/2410.08020)), and we explore this direction. In Appendix C, we demonstrate that MSA-based ProteinTTT further improves fitness prediction compared to single-sequence ProteinTTT (at the cost of MSA construction). Inspired by the reviewer’s suggestion, we additionally apply ProteinTTT$_{\text{MSA}}$ to ESMFold for structure prediction. We find that **ESMFold \+ ProteinTTT is further improved when combined with MSA**. Table R2 shows substantially improved pLDDT on 100 random proteins from BFVD (we will extend this analysis to the full database, for the next revision). Please note that pLDDT upon ProteinTTT correlates well with downstream LDDT (Figure A9).
> >
> > Table R2. Structure prediction performance on 100 random proteins from BFVD. The table reports the average pLDDT. For efficiency during the rebuttal, we use only a single random seed.
> >
> > | Method | pLDDT ↑ |
> > | :---- | :---- |
> > | ESMFold | 53.7 |
> > | ESMFold+ProteinTTT | 66.0 |
> > | ESMFold+ProteinTTT$\_\\text{MSA}$ | 80.42 |
> >
> > Beyond using MSAs, future research could combine ProteinTTT with retrieval-augmented (RAG) inference, inspired by RAG-style models ([Weitzman et al., 2025](https://arxiv.org/abs/2506.08954), [Lewis et al., 2020](https://arxiv.org/abs/2005.11401), [Shaw et al., 2024](https://www.biorxiv.org/content/10.1101/2024.05.30.596539v2)) or nearest-neighbor inference ([Khandelwal et al., 2020](https://arxiv.org/abs/1911.00172)), which could supply additional structural or functional context at prediction time. Lightweight inference-time ensembling or mixture-of-experts routing ([Fedus et al., 2022](https://arxiv.org/abs/2101.03961)) may also help adapt predictions to local sequence neighborhoods in a manner complementary to gradient updates. Additionally, iterative inference procedures analogous to self-consistency decoding ([Hayes et al., 2025](https://www.science.org/doi/10.1126/science.ads0018), [Wang et al., 2023](https://arxiv.org/abs/2203.11171)) could further enhance ProteinTTT (we evaluate chain of thought with ESM3 in Table 1, where ProteinTTT performs better). Beyond these directions, ProteinTTT can be combined with mechanistic interpretability approaches for improved controllability and analysis ([Hübotter et al., 2025](https://arxiv.org/pdf/2509.24510)). We will extend the text to clarify that **ProteinTTT could benefit from these inference-time scaling strategies as well**.
> >
> > > Overall, While the work is technically sound and relevant, its contributions appear more empirical and domain-specific than methodological. The paper might be more impactful if expanded with additional biological validations. In its current form, the scope and framing could align more naturally with a computational biology or bioinformatics venue rather than a general ML conference.
> >
> > We believe that presenting ProteinTTT at a general machine learning conference could have an important impact on the further development of machine learning applied to biology. First, to the best of our knowledge, **ProteinTTT effectively opens up the field of test-time approaches in computational biology and would benefit from further investigation by the machine learning community**. This is particularly timely, as generalization to out-of-distribution examples remains a central issue in machine learning for biology. Second, our method is broadly applicable across major subdomains of machine learning applied to biology, such as protein structure, fitness, and function prediction, and **can be adapted and extended by machine learning researchers for specialized methods** in these tasks.

---

> > > ### Comment · Reviewer_DpSs · 2025-11-20
> > >
> > > I really appreciate the response. I raised the contribution to 3 and overall score to 6.

---

### Author Response · Authors · 2025-11-20

We thank all the reviewers for their thorough and high-quality reviews. We will provide clarifications and conduct additional experiments to address each of the raised points. Responses will be posted one by one as soon as they are ready.

---

### Author Response · Authors · 2025-12-01

We thank all reviewers for the time and effort invested in their careful reviews and in the constructive discussion. We have addressed each of the points raised and have now updated the PDF with all changes marked in blue.

Our work proposes ProteinTTT, to the best of our knowledge, the first test-time customization method in biological machine learning. We summarize the main discussion points below.

**Strengths**

- **Clarity**. Reviewers found the paper “well written, clear, and easy to follow” (9CBZ) and “exceptionally clearly written” (MtJ2).
- **Novelty**. Reviewers noted that the paper introduces “a novel and insightful perspective” (9CBZ), that “the idea of introducing test-time training to PLMs is interesting and convincing” (wNTv), and that this is “the first paper… that applies this sort of idea to protein modeling” (MtJ2).
- **Applicability**. Reviewers mentioned that the method is “simple and general… does not require additional data and can be easily applied to different pretrained PLMs” (DpSs) and “transplantable across diverse backbones” (wNTv). They also noted its “lightweight finetuning… making it resource-efficient” (DpSs).
- **Validation**. Empirical validation was described as “comprehensive, covering three relevant downstream tasks” (9CBZ), “rigorously conducted” (wNTv), and producing “consistent” (DpSs) and “convincing” results (9CBZ) on a “diverse set of experiments” (MtJ2). Reviewers noted that “most proteins can be improved by ProteinTTT” (wNTv) and appreciated that “case studies demonstrate real-world applications” (DpSs) and “further strengthen the empirical validation” (9CBZ).

**Weaknesses (and how we addressed them)**

- **Applicability beyond masked language modeling**. Reviewers noted that ProteinTTT “seems inherently limited… to sequence-based tasks” (9CBZ) and questioned the lack of autoregressive (DpSs) or diffusion models (9CBZ).
  We implemented ProteinTTT for autoregressive ProGen2-small and ProGen2-large, as well as for the discrete-diffusion DPLM-BiT model (Tables 1, 2), both showing significant improvements. We also added a new section “Extension to other model types and tasks” in Appendix I explaining even broader applicability.
- **Magnitude of improvements**. Reviewers observed that “details about the magnitude of the improvement … seem to be pretty sparse” (MtJ2) and may appear “marginal” (DpSs, 9CBZ).
  We clarified that the improvements are statistically significant and added large-scale improvement distribution on protein structure prediction (new Fig. A12) showing substantial per-protein gains.
- **Limitations and failure modes**. Reviewers suggested “deeper analysis of cases where ProteinTTT fails” (9CBZ, wNTv).
  We added a section discussing limitations (new Appendix I “Limitations and future work”) and performed new experiments to show that failure cases largely stem from ambiguous ground truth (new Fig. A13) or high-MSA-depth proteins, well-represented in training data (Appendix H). We also quantified improvement rates across all models (Tables R4–R5), confirming that the majority of samples consistently show improvement.

For convenience, we summarize below the final reviewer scores at the end of the discussion period. Please note that the score upgrades in the discussion took place on Nov 20 and Nov 25, both prior to the deanonymization leak on Nov 27\.

Reviewer DpSs: 6 ("I really appreciate the response. I raised the contribution to 3 and overall score to 6.")

Reviewer 9CBZ: 8

Reviewer MtJ2: 6 ("This all looks good to me. Raising from (4 \-\> 6).")

Reviewer wNTv: 6 ("Seems fine to me. Most proteins can be improved by ProteinTTT. I decide to keep the score.")

---

### Meta-Review · Area_Chair_K599 · 2026-01-03

**Summary:**

This paper focuses on the generation problem of protein learning, the authors introduce protein test-time training method that customizes protein language model backbones on individual sequences during inference to improve generalization. The experiments show strong improvements.

While reviewers unanimously praised the paper's clarity and the intuitive appeal of applying test-time adaptation to biology, initial concerns focused on the magnitude of performance gains and the restriction to masked language modeling architectures. The consensus for acceptance was solidified after the authors demonstrated the method's versatility across autoregressive and diffusion models and provided rigorous statistical evidence of performance improvements.

**Reviewer Concerns:**

The authors addressed the primary critique regarding the method's scope by implementing ProteinTTT on ProGen2 (autoregressive) and DPLM-BiT (discrete diffusion), proving the approach extends beyond masked language modeling.
Concerns regarding "marginal" gains on large models were effectively rebutted with new distribution plots showing substantial improvements for out-of-distribution sequences and confirming statistical significance across benchmarks.
The request for failure analysis was also resolved by demonstrating that performance degradation primarily occurs in cases of ambiguous ground truth, such as unmodeled dimers or NMR ensembles, rather than algorithmic failure.

**Reviewer Scores:**

The comprehensive rebuttal led to a distinct upward trend in scoring, with Reviewers DpSs and MtJ2 both raising their scores from 4 to 6 after seeing the expanded model support and "oracle" experiments validating the pLDDT stopping criteria. Reviewer wNTv maintained a positive score of 6, expressing satisfaction with the analysis of improved sample percentages, while Reviewer 9CBZ retained a strong score of 8. All reviewers now actively support acceptance, reflecting a robust consensus on the paper's contribution to biological machine learning.

---

### Decision · Program_Chairs · 2026-01-26

Accept (Poster)